# The diurnal cycle of the $p$CO$_2$ in the coastal region of the Baltic Sea

Martti Honkanen[1], Jens Daniel Müller[2,4], Jukka Seppälä[3], Gregor Rehder[4], Sami Kielosto[1,3], Pasi Ylöstalo[3], Timo Mäkelä[5], Juha Hatakka[5], and Lauri Laakso[1,6]

[1]Meteorological and Marine Research Programme, Finnish Meteorological Institute, Finland
[2]Environmental Physics, Institute of Biogeochemistry and Pollutant Dynamics, ETH Zurich, Zurich, Switzerland
[3]Marine Research Center, Finnish Environment Institute, Finland
[4]Department of Marine Chemistry, Leibniz Institute for Baltic Sea Research, Warnemunde, Germany
[5]Climate Research Programme, Finnish Meteorological Institute, Finland
[6]School of Physical and Chemical Sciences, North-West University, Potchefstroom Campus, South Africa

**Correspondence:** Martti Honkanen (martti.honkanen@fmi.fi)

**Abstract.** The direction and magnitude of carbon dioxide fluxes between the atmosphere and the sea are regulated by the gradient in the partial pressure of carbon dioxide ($p$CO$_2$) across the air–sea interface. Typically, observations of $p$CO$_2$ at the sea surface are carried out by using research vessels and Voluntary Observing Ships, which usually do not resolve the diurnal cycle of $p$CO$_2$ at a given location. This study evaluates the magnitude and driving processes of the diurnal cycle of $p$CO$_2$ in a coastal region of the Baltic Sea. We present $p$CO$_2$ data from July 2018 to June 2019 measured in the vicinity of the island of Utö at the outer edge of the Archipelago Sea, and quantify the relevant physical, biological, and chemical processes controlling $p$CO$_2$. The highest monthly median of diurnal $p$CO$_2$ variability (31 µatm) was observed in August and predominantly driven by biological processes. Biological fixation and mineralisation of carbon generated sinusoidal diurnal $p$CO$_2$ variations, with maxima in the morning and a minima in the afternoon. Compared with the biological carbon transformations, the impact of air–sea fluxes and temperature changes on $p$CO$_2$ were small, with their contributions to the monthly medians of diurnal $p$CO$_2$ variability being up to 12 and 5 µatm, respectively. During upwelling events, short-term $p$CO$_2$ variability (up to 500 µatm within a day) largely exceeded the usual diurnal cycle. If the net annual air–sea flux of carbon dioxide at our study site and for the sampled period is calculated based on a data subset that consists of only one regular measurement per day, the bias in the net exchange depends on the sampling time and can amount up to ±12%. This finding highlights the importance of continuous surface $p$CO$_2$ measurements at fixed locations for the assessment of the short-term variability of the carbonate system and the correct determination of air–sea CO$_2$ fluxes.

## 1 Introduction

Over the last decade (2009–2018), anthropogenic carbon dioxide (CO$_2$) emissions to the atmosphere amounted to 11 gigatonnes carbon per year, mainly driven by fossil fuel combustion, land use change, and cement production. Approximately half

of these emissions were bound by the terrestrial biosphere ($3.2\,\mathrm{GtCy^{-1}}$) and the oceans ($2.5\,\mathrm{GtCy^{-1}}$) together (Friedlingstein et al., 2019). The increased $CO_2$ concentration in the atmosphere causes global warming, while the increase of $CO_2$ dissolved in the oceans drives ocean acidification (Feely et al., 2009). The correct quantification of air–sea fluxes of $CO_2$ is thus an essential component to keep track of the redistribution of anthropogenic carbon within the earth system and asses its potentially

harmful impact. The air–sea $CO_2$ fluxes can undergo large daily variations, and thus it is vital to understand the daily dynamics of the processes driving the flux in order to provide accurate estimate of the net annual air–sea $CO_2$ fluxes.

The partial pressure of $CO_2$ ($pCO_2$) in surface seawater and thus the direction of the air–sea $CO_2$ flux ($F_{as}$) are mainly regulated by the interplay of biological productivity and respiration, temperature-induced changes in seawater carbonate chemistry, and mixing processes. As the sea surface receives more solar radiation during the day than at night, a diurnal cycle in the

biology, physics, and chemistry of the surface seawater establishes. Since sea surface $pCO_2$ observations are widely used for calculating the $CO_2$ exchange between the sea and the atmosphere, there can be large discrepancies between the flux estimates when using $pCO_2$ values measured at different times of the day.

The diurnal variation of the $pCO_2$ is typically larger in coastal seas than in the open oceans (Goyet and Peltzer, 1997) due to the larger biological activity. The diurnal $pCO_2$ cycle has been studied e.g. in an oligotrophic ocean (Olsen et al., 2004), at coral

reefs (Yan et al., 2016), and in tidal regions (Andersson and Mackenzie, 2012). A limited number of studies have addressed the diurnal cycle of $pCO_2$ in the Baltic Sea. Lansø et al. (2017) found that there was no evident diurnal $pCO_2$ signal in the Baltic Proper and Arkona Basin in winter time, but during April–October, the monthly average $pCO_2$ amplitudes were up to $27\,\mathrm{\mu atm}$. Wesslander et al. (2011) determined that the diurnal $pCO_2$ variability in the Baltic Proper was mainly controlled by biological processes, mixing, or the air–sea exchange of $CO_2$.. Huge (up to $1604\,\mathrm{\mu atm}$) diurnal variability of $pCO_2$ in a highly

productive macrophyte meadow in the Western Baltic Sea was reported by Saderne et al. (2013).

The carbon system of the Baltic Sea shows large spatial variability. On the one hand, the northern part of the Baltic Sea, i.e., the Gulf of Bothnia, is characterized by large fluvial fluxes of organic matter into its basins, which turns the area into a source of $CO_2$ for the atmosphere through effective bacterial remineralization (Algesten et al., 2006). On the other hand, the southern parts of the Baltic Sea exhibit larger primary production compared with the Gulf of Bothnia (Wasmund et al., 2001),

a larger input of alkalinity from land, and lower input of organic matter, which makes the basin act as a carbon sink (Kuliński and Pempkowiak, 2011). Based on the mass balance approach of Kuliński and Pempkowiak (2011), revisited by Ylöstalo et al. (2016), the Baltic Sea as a whole is considered to be a weak source of carbon dioxide for the atmosphere.

Measurements of $pCO_2$ taken by Ships of Opportunity (SOOPs) have proved to be a cost-effective method to reveal new insights into the spatio-temporal variability of the Baltic Sea's carbon cycle (Schneider et al., 2014; Schneider and Müller, 2018).

These surface $pCO_2$ measurements carried out on SOOP routes are currently our best presentation of the spatial variability of $CO_2$ partial pressure in the Baltic Sea. However, the measurements carried out on these fixed routes and time schedules do not resolve the diurnal cycle, and when interpreting these data, one should consider the potential bias caused by the time of the sampling. Fixed stationary platforms, though limited in their spatial coverage, are capable of measuring in high temporal resolution resolving the diurnal cycle of $pCO_2$, and thus provide data highly complementary to data retrieved on SOOPs or

RVs. .

In this contribution, we investigate the diurnal cycle of carbon dioxide system at a fixed station near the island of Utö, located in the transition zone between the northern Baltic Proper and the Archipelago Sea, representing a highly productive (euthrophied) coastal ecosystem. The aims of this study are (a) to investigate the diurnal cycle of $p\text{CO}_2$ during different seasons based on observations carried out at Utö and (b) to quantify the contributions of the main drivers and processes affecting the
$p\text{CO}_2$ diurnal variations: air–sea flux, biological carbon uptake and release, and diurnal changes in temperature.

## 2  Materials and methods

### 2.1  Study site

The Utö Atmospheric and Marine Research Station is located on the island of Utö (Fig. 1) on the southern edge of the Archipelago Sea (59°46'55" N, 21°21'27" E). Utö is a small (0.81 km$^2$) rocky island with low vegetation.

As characteristic for the central Baltic Sea, our study site is affected by climate change induced increase of sea water temperature (Laakso et al., 2018). Besides the warming trend, also stratification has strengthened, affecting the connectivity between water layers separated by a seasonal thermocline and a permanent halocline (Liblik and Lips, 2019). Long-term trends of increasing alkalinity throughout the Baltic Sea have been shown to partly compensateacidification induced by rising atmospheric $\text{CO}_2$. (Müller et al., 2016). Within our study region, phytoplankton blooms are a recurrent phenomenon due to
eutrophication (Kraft et al., 2021).

The marine observations at the station focus on regional marine ecosystem functioning with a large number of biochemical and physical observations. The marine observations include, but are not limited to, CTD casts carried out northwest from the island, flow-through analyses at the Marine station and thermistor measurements in the vicinity of the seawater inlet (Fig 1). The measurements of the Utö Atmospheric and Marine Research Station belong to the Joint European Research
Infrastructure for Coastal Observatories (http://www.jerico-ri.eu). Carbonate system dynamics is noted as one of the key scientific topics in coastal ocean studies (Farcy et al., 2019), and the study presented here, executed under the framework of the JERICO-RI, highlights the need for integrated and multidisciplinary observations. The atmospheric part of the station includes a wide range of meteorological, greenhouse gas and aerosol measurements. The micro-meteorological flux tower at the western shore, next to the Marine station, measures the $\text{CO}_2$, sensible heat and latent heat fluxes between the sea and the atmosphere.
Greenhouse gas monitoring and some meteorological measurements are part of the Integrated Carbon Observation System Research Infrastructure (ICOS RI). For the complete list of observations, visit the Finnish Meteorological Institute's web site (https://en.ilmatieteenlaitos.fi/uto-observations). Site bathymetry and other information about the study site are given in Laakso et al. (2018) and Kraft et al. (2021). Our study is based on one year's data gathered between July 2018 and July 2019. The timing of all data presented in this paper are given in UTC. Finland belongs to the UTC+2:00 timezone.

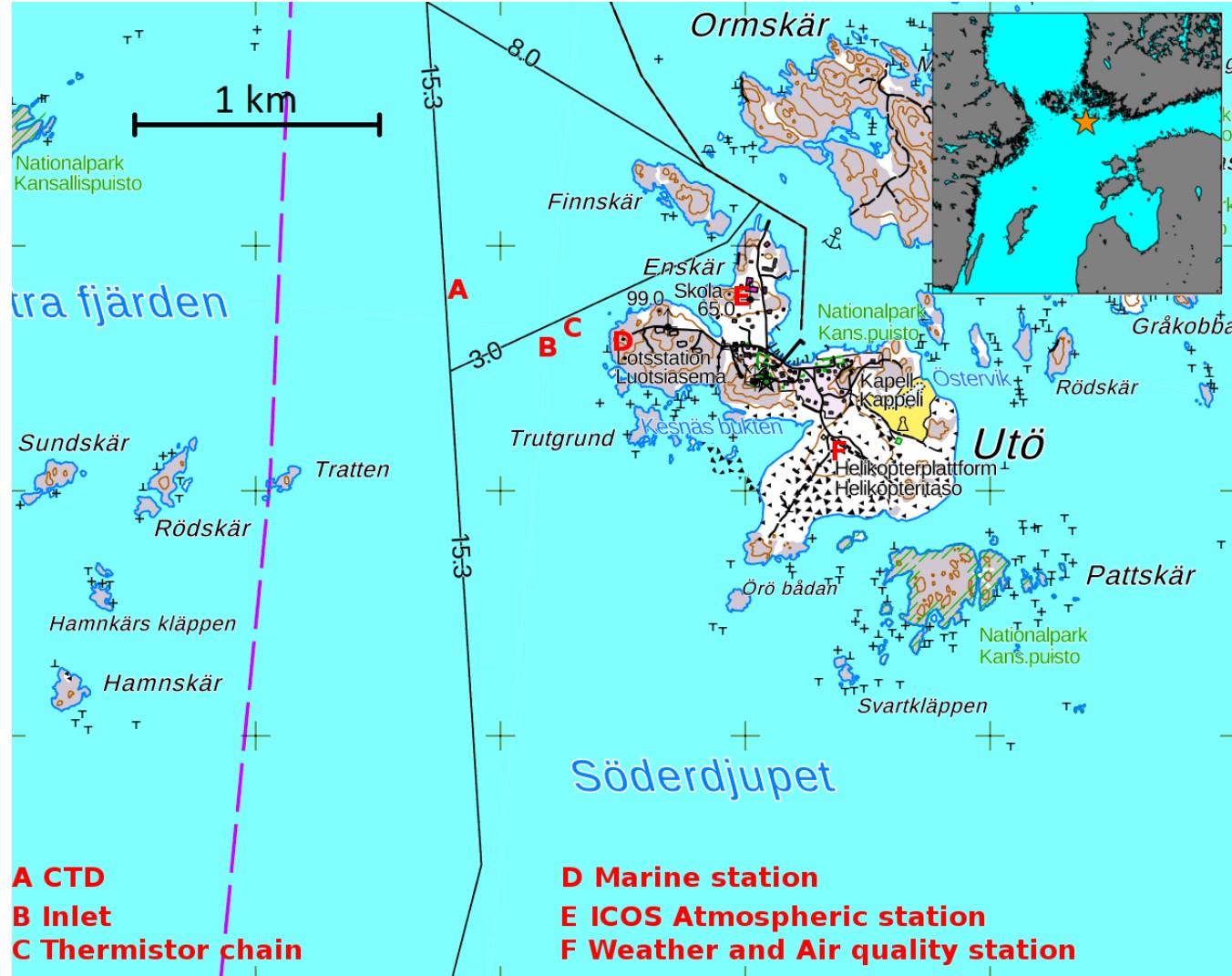

**A CTD**
**B Inlet**
**C Thermistor chain**
**D Marine station**
**E ICOS Atmospheric station**
**F Weather and Air quality station**

**Figure 1.** Sampling locations at Utö Atmospheric and Marine Research Station. The grid size (distance between plus signs) is 1 km. The smaller figure on the upper right corner shows the location of Utö (orange star). The National Land Survey of Finland is acknowledged for providing the map.

## 2.2 Flow-through sampling

The marine station, located on the western shore of the island (Fig. 1), is equipped with a flow-through system. A submersible pump located 250 m from the shore transports seawater from the inlet to the marine station, where seawater is analyzed automatically or manually on demand. The bottom-moored floating seawater inlet is at the approximate depth of 4.5 m $\pm$ 0.5 m. The mean depth at this location is 23 m and the sea level at Utö varies $\pm$0.5 m relative to theoretical mean sea level. At the location, there are no notable tides or tidal currents.

At the station, the transported water first enters a manifold. Any flow-through instrument can be attached to the manifold separately, enabling individual adjustment of the flow rate for each instrument. The time stamp of the flow-through data is shifted (5.6min on average) according to the concurrent flow rate (54–68 LPM) to match the time of sampling at the intake, based on the known volume of the pipe system.

All of the instruments attached to the flow-through system are automatically washed with cleaning fluid (hydrogen peroxide or Triton X-100) daily. The data gathered during and immediately after the cleaning have been discarded.

### 2.2.1   Measurement of $p\text{CO}_2$

A SuperCO$_2$ instrument (Sunburst Sensors), which was connected to the flow-through system, was used to measure $p\text{CO}_2$. In its two shower-head equilibrator chambers, the seawater CO$_2$ is equilibrated with the gas above according to Henry's law (Eq.

B2). The equilibrated gas is analysed for its CO$_2$ molar fraction ($x\text{CO}_2$) by an infrared gas analyzer (LI-840A, LI-COR). The logging interval was 10–15 s.

     The sensor drift of the gas analyzer is taken into account by measuring four standard gases every fourth hour with differing CO$_2$ molar fractions (0.00, 234.38, 396.69, and 993.45 ppm, $\pm 2\%$) in order to form a correction equation for dry $x\text{CO}_2$. FMI buys the reference gases from the Finnish branch of Linde-Gas (previously AGA). The gas concentrations are checked with

instruments using cavity ring-down spectroscopy in the FMI's laboratory prior to measurements. These instruments are calibrated using gases that are verified by the National Oceanic and Atmospheric Administration (USA). Aluminum gas containers have been used in order to minimize the concentration drift.

     Drift-corrected dry $x\text{CO}_2$ is transformed into $p\text{CO}_2$ as described in Dickson et al. (2007), with a slight modification. Since the water trap attached to the sample gas line may slightly affect the water vapor content, the following calculation was used.

The dry CO$_2$ molar fraction was calculated using the H$_2$O measured using the analyzer. The real water vapor content in the equilibrium chambers was calculated using the temperature and salinity data assuming full saturation. This real water vapor content, together with the dry $x\text{CO}_2$, was used when calculating the partial pressure of CO$_2$.

     During May–June 2019, the sampling and inlet tube system was tested by measuring $p\text{CO}_2$ with two SAMI[2] sensors (Sunburst Sensors) that were parallel to the SuperCO$_2$ system inside the measurement station on land (20–23 May 2019), followed

by deployment of the SAMI[2] sensors next to the sampling inlet at sea (from 24 May to 7 June 2019). The parallel measurement inside the station was used to correct the potential initial offset of the SAMI[2] sensors against the SuperCO$_2$ system. While the SAMI[2] sensors were positioned close to the inlet at sea, the in-situ concentrations for all three instruments closely followed each other: the root mean square difference between measurements at the sea inlet and the station was 4.1 µatm. We conclude that the $p\text{CO}_2$ analysis carried out in the station, despite the unusal long path of water from the inlet location to the lab, fully

represent the conditions at the inlet.

### 2.2.2   Other flow-through measurements

The equilibrator temperature (together with salinity) was measured using a thermosalinograph (SBE45 MicroTSG, Sea-bird Scientific) next to the SuperCO$_2$ instrument. The thermosalinograph is cleaned 1-2 times a year. The accuracies for temperature

and salinity given by the manufacturer are respectively 0.002 °C and 0.005. The temperature drift is less than a few thousandths of a degree per year, whereas the stability of conductivity measurement depends mostly on the cleanliness of the measurement cell. The thermosalinograph logged data every 15 s.

Oxygen was measured with an oxygen optode (Aanderaa 4330) with multipoint calibration. The optode has a preburned foil providing long term stability. The accuracy of the optode is 2 μM according to the manufacturer. For the work presented here, we are mostly interested in hourly changes of oxygen, and thus the drift of the absolute value is not concern. Chlorophyll A was measured with a Wetlabs FLNTU fluorometer, as a proxy of chlorophyll concentration, using factory calibration. Both were connected to the flow-through system. Chlorophyll A measurement was offline in winter (January–March). Both instruments logged data every 15 s.

## 2.3 Measurements from other sampling locations

### 2.3.1 Hydrographic measurements

The vertical temperature profiles were measured with temperature chains, supported with regular interval profiles of Conductivity–Temperature–Depth instrument (CTD), RBR XR-620. The CTD profiles were taken fortnightly by using a small boat during the productive period and with lower temporal resolution in winter (see Fig. 2). The CTD location is approximately 400 m west of the sampling inlet.

The thermistor chain was deployed 150 m northeast from the seawater inlet in July 2018; this chain was moored at the depth of 21.3 m $\pm$ 0.5 m, and its Pt-100 thermistors were placed at the heights of approximately 18 m, 13 m, 8 m, 1 m, and 0 m from the bottom (depths 3.3 m $\pm$ 0.5 m, 8.3 m $\pm$ 0.5 m, 13.3 m $\pm$ 0.5 m, 20.3 m $\pm$ 0.5 m, 21.3 m $\pm$ 0.5 m). In order to avoid instrument damages during rough weather conditions, there was no thermistors closer than 3 m to the surface. Pt-100 thermistors were calibrated prior to the deployment in FMI's laboratory, and the maximum error in temperature was found to be less than 0.015 °C. Thermistors logged data every 30 s.

The thermistor profiles were used to verify that the CTD casts, carried out at a slightly different location, were representative for the hydrographic conditions at the seawater inlet. More importantly, the 3 m thermistor measurement was used as insitu temperature at the inlet, and hence for correcting the $pCO_2$ for the temperature difference between in situ conditions and in the equilibration chamber.

### 2.3.2 Atmospheric $CO_2$ measurement

The atmospheric $xCO_2$ was measured at the Atmospheric ICOS site. The sample air was drawn from the tower (56 m) to the ground level where it was analyzed using using cavity ring-down spectroscopy (Picarro G2401). The data was logged as one minute average values. Three standard gases made by FMI were used for the reference measurement. Differences between the target and measured values of these gases were within -0.20 and 0.20 ppm.(Kilkki et al., 2015)

### 2.4 Calculated data

#### 2.4.1 $pCO_2$ temperature correction

To correct for the temperature difference between in situ and equilibrator temperature, we took the effect of the temperature change on $pCO_2$ into account by using the CO2SYS matlab program (van Heuven et al., 2011). This correction requires knowl-
edge of another carbon system component, which is total alkalinity (from salinity) in our case. The widely used temperature correction of $pCO_2$ suggested by Takahashi et al. (1993) is not applicable for the brackish conditions of the Baltic Sea (e.g. Schneider and Müller, 2018). The difference in temperature oscillates within $\pm 2.0\,^\circ\mathrm{C}$.

#### 2.4.2 Determination of the mixed layer depth

The mixed layer depth ($z_{\mathrm{mix}}$) was determined from the vertical temperature profiles of the CTD casts. Even though the data
by the thermistor chain has higher temporal resolution than the CTD castings, it is not applied for the assessment of the mixed layer depth because it has significantly lower vertical spatial resolution. The water depth at the location of CTD casts is approximately $90\,\mathrm{m}$, which is significantly deeper than the depth at the inlet location. If the mixed layer depth was deeper than the depth of 23 m at the inlet location, the water column at the inlet location was considered fully mixed. The thermocline depth, i.e., the depth of the strongest temperature gradient in the profile, was considered to represent $z_{\mathrm{mix}}$. For each CTD cast, a
thermocline depth was estimated. The thermocline depths with a questionably small ($< 0.2\,^\circ\mathrm{C}\,\mathrm{m}^{-1}$) temperature gradient were discarded.

Due to the marked horizontal distance between the inlet and CTD profiling, the applicability was assessed by comparing these CTD measurements to the Pt-100 thermistor chain measurements near the inlet, which confirmed the relatively good match of the measurements with the root mean square difference of $0.6\,^\circ\mathrm{C}$. The CTD measurements reproduced well the
hydrography of the upper water column at the inlet location, as the root mean square differences between the sites for the depths of 3, 8, and $13\,\mathrm{m}$ were 0.42, 0.41, and $0.25\,^\circ\mathrm{C}$, respectively. The temperatures at 20 m, however, showed larger difference as the root mean square error (RMSE) was $1.08\,^\circ\mathrm{C}$ for this depth. This implies that the mixed layer depths were well reproduced using the CTD castings unless the thermocline was located close to the bottom of the inlet location.

#### 2.4.3 Estimation of $F_{as}$

The estimation of the air–sea exchange of $CO_2$ between the sea and atmosphere used in this study is based on two methods: (1) the eddy covariance method, using the data gathered using a micro-meteorological flux tower erected on the western shore of the island and (2) a wind speed-based flux parameterization. Due to strict quality control, the eddy covariance method was applicable for only 18% of time, and for the rest of the time, the parameterization was used.

Both methods have pros and cons, due to which they complement each other. The eddy covariance method considers the
integrated flux within a large footprint area, whereas the parameterization is based on the $pCO_2$ measurement at a single point at the depth of $4.5\,\mathrm{m}$. The large footprint area may contain spatially heterogeneity in seawater $pCO_2$. In some cases, the

measurement at the depth of 4.5 m may not represent the surface conditions. Additionally, the parameterization of gas transfer velocity is based on the wind speed, which does not contain all the information about the surface turbulence used alone, in particular close to land masses.

The eddy covariance fluxes for the air–sea exchange of $CO_2$ were calculated at 30 min intervals. This flux measurement is based on the closed-path non-dispersive infrared gas analyzer (LI-7000, LI-COR). The sample air tubing has a 30 cm Nafion drier (PD-100T-12-MKA, Perma Pure) in order to eliminate the water vapor interference of $CO_2$ fluxes. The covariance of 10 Hz vertical wind velocity ($w$) and $CO_2$ molar fraction ($x CO_2$) data was calculated for each 30 min averaging period. These fluxes were corrected for the high-frequency attenuation by using a transfer function that was calculated from the deviation of the normalized $w$-$CO_2$ cospectrum from the cospectrum of sensible heat flux. Only stationary $CO_2$ flux conditions were included because, during non-stationary conditions, the measured fluxes do not represent the exchange between the surface and the atmosphere. Only westerly winds were considered (180–330 °) here as the flux footprint during these cases originates from the sea. A small amount of flux data were excluded from the analysis because the reference gas pipeline for the $CO_2$ analyzer was leaking. More information about the flux system and its quality control can be found in Honkanen et al. (2018).

We used an air–sea exchange estimation based on the quadratic relationship created by Wanninkhof (2014) for the times without valid eddy flux measurements (82% of the time). Wind speed was measured with the micrometeorological flux tower on the western shore, and data were converted to wind speed at the height of 10 m, $U_{10}$. As the wind speed is not precisely measured at the height of 10 m, we corrected wind speed assuming a logarithmic wind profile and a constant surface roughness of 0.5 mm, an average value that is based on the data of Honkanen et al. (2018). More details about the compatibility of the parameterization for this specific site can be found in Appendix A1.

### 2.4.4 Alkalinity–salinity relationship

We use total alkalinity as a second carbon system variable in our calculations. The total alkalinity used here is calculated using the alkalinity–salinity relationship:

$$TA(\text{µmol kg}^{-1}) = 123.3 + 221.8 \cdot S, \tag{1}$$

where salinity is unitless and total alkalinity has the unit of µmol kg$^{-1}$. This is based on the samples gathered from the flow-through system at Utö in summer 2017 (Lehto, 2019). Total alkalinity was determined from these samples by using the potentiometric titration method (Metrohm Titrino 716). The samples were conserved with mercury chloride before the analysis in Finnish Environment Institute's research laboratory in Helsinki. The titrant and the rinsing water had the salinity of 7. Alkalinity was calculated from the titration curve based on the least squares method. More information on the alkalinity–salinity relationship, can be found in Appendix C.

### 2.4.5 The calculation of the $p$CO$_2$ changes generated by different processes

The surface $p$CO$_2$ is affected by processes that change the concentrations of dissolved inorganic carbon (DIC) or total alkalinity (TA), or through changes in temperature, salinity, or pressure affecting the carbonate system balance (Takahashi et al., 1993).

In contrast to $pCO_2$, DIC and TA behave conservative with respect to temperature changes and mixing of water masses, when expressed in concentration units of $\mu mol\,kg^{-1}$ of seawater.

As DIC (see Appendix B) is introduced to or removed from the dissolved inorganic pool, its change is depicted by the so-called Revelle factor, $Re$ (Sarmiento and Gruber, 2004):

$$Re = \frac{\Delta[CO_2]}{[CO_2]} \bigg/ \frac{\Delta DIC}{DIC}. \tag{2}$$

DIC in surface water is affected by the $CO_2$ exchange with the atmosphere, biological transformations, precipitation/dissolution of calcium carbonate, fresh water input, and the mixing of water masses. The processes controlling the freshwater balance include evaporation, precipitation and the formation and melting of sea ice. Precipitated water or melted sea ice may produce a layer of low saline water at the sea surface, which in most cases is likely to be eroded easily by turbulence.

Biological processes affecting $pCO_2$ include all transformations between the inorganic and organic carbon pools, i.e., photosynthesis and respiration. The mixing processes include horizontal advection, vertical diffusion, and vertical entrainment.

TA (see Appendix C) is mainly altered by the formation and dissolution of calcium carbonate. A smaller contribution to TA originates from nitrogen transformations through biological processes, and the mixing processes. TA is not affected by the air–sea exchange of $CO_2$. The effect of calcifying primary producers in the carbon pool can be neglected for the open Baltic Sea (Tyrrell et al., 2008). However, calcifiers may have an effect on the carbon cycle in the benthic zone.

Temperature affects the dissociation constants and solubility of gases, which further alters the $CO_2$ partial pressure. For stable oceanic conditions, this change is well documented (Takahashi et al., 1993), but in estuary conditions, the temperature effect on $pCO_2$ varies significantly (Schneider and Müller, 2018). Based on the choice of the parameterization of dissociation constants, this value might show small variation as a function of temperature and salinity (Orr et al., 2015). Similarly to temperature, salinity and pressure also affect the dissociation constants.

In this study, we investigate the contribution of individual processes and drivers to the diurnal variation of $pCO_2$. We are considering the $pCO_2$ changes that are generated by the changes in DIC or by temperature fluctuations. DIC changes are further divided into the changes that are caused by the air–sea exchange of $CO_2$ or by biological transformations. There are multiple other processes that have the potential to affect the $pCO_2$ that are not included in the analysis. See Appendix C1 for more information on the omitted processes.

Calculations of the carbon system were performed using the CO2SYS matlab program (van Heuven et al., 2011). Dissociation constants $K_1$ and $K_2$ were calculated based on the work of Millero (2010) and the sulfate contribution is based on the work of Dickson et al. (2007). We implemented the total boron parameterization of Kuliński et al. (2018), which is based on the empirical data of the Baltic Sea, in CO2SYS.

First, the carbon chemistry is calculated in CO2SYS for each hour based on the measured partial pressure of $CO_2$, parameterized total alkalinity (see above), temperature and salinity. This results in hourly data of DIC at the sea surface.

In the case of the hourly temperature-related $pCO_2$ change, we assume that DIC and TA do not change. Using the temperature of the next hour together with the previously known DIC and TA, we calculate the new $pCO_2$ in CO2SYS that is governed by solely the temperature change

In the case of air-sea exchange and biological transformations, we calculate how much DIC has changed over one hour by these processes separately and add this DIC change, dDIC, to the original DIC content. Then, we calculated the carbon system using this new DIC and the unaltered total alkalinity in order to get the new $pCO_2$.

We assume that the new inorganic carbon ($dDIC_A$) derived from the air–sea exchange of carbon dioxide is evenly dis-
tributed within the mixed layer. The DIC change due to the air–sea exchange of $CO_2$ is calculated as:

$$dDIC_A = \frac{F_{as}}{z_{mix}}\Delta t \tag{3}$$

where $t$ is time, in our case one hour. The value of $F_{as}$ is calculated using either the eddy covariance method or the wind speed-based parameterization, with the former given priority when passing our rigorous quality control procedure (18% of the time considered in this study).

We inferred the biological effect on DIC indirectly from the oxygen measurements by assuming the Redfield ratio (Redfield et al., 1963). As inorganic carbon is consumed (or released), a corresponding amount of oxygen is released (or consumed):

$$dDIC_B = -\frac{106}{138}\Delta[O_2] - \frac{FO_2}{z_{mix}}\Delta t \tag{4}$$

The ratio of 106 C : -138 O refers to the Redfield ratio of carbon to oxygen (Redfield et al., 1963). However, this ratio is based on average oceanic conditions and may show variations in space and time. The last term in the equation takes the effect
of air–sea exchange of oxygen into account. This flux, $FO_2$, is calculated similarly to the carbon dioxide flux (Eq. A1) by using the gas transfer velocity and the oxygen solubility, the measured oxygen concentration in seawater, and the oxygen concentration calculated for hypothetical equilibrium with the atmosphere. Oxygen solubility was calculated according to the salinity-temperature dependence fit of Garcia and Gordon (1992), which is originally based on the work of Benson and Krause (1980). The Schmidt number of oxygen and gas transfer velocity were calculated according to Wanninkhof (2014). Oxygen
concentrations can also change due to mixing, the contribution of which remains unknown.

For each day, the cumulative sums of the hourly $pCO_2$ changes generated by a specific process (temperature, biological transformations or air-sea exchange of $CO_2$) were calculated for 00:00 – 24:00, in order to know how the specific process alters the $pCO_2$ during a day. Finally, the mean of cumulative sum was removed from these values, because we are interested in the daily changes, not the absolute values. $pCO'_{2,i}$ is the cumulative pCO2 change between the i:th and the first hour:

$$pCO'_{2,i} = \sum_{i=1}^{24}\Delta pCO_{2,i} - \left< \sum_{i=1}^{24}\Delta pCO_{2,i} \right> \tag{5}$$

where $i$ is the index of each hour and the angle brackets denote the averaging.

In addition to the $pCO_2$ evolution generated by the air–sea exchange of $CO_2$, biological transformations, and temperature alone, we also examined the $pCO_2$ evolution generated by these three processes simultaneously. This is calculated using the DIC that is altered by both the air-sea exchange of $CO_2$ and biological transformation, and additionally taking into account the
temperature change. However, this $pCO_2$ change is only used for the verification of the method, and as base for the discussion of the shortcomings and potential improvements. .

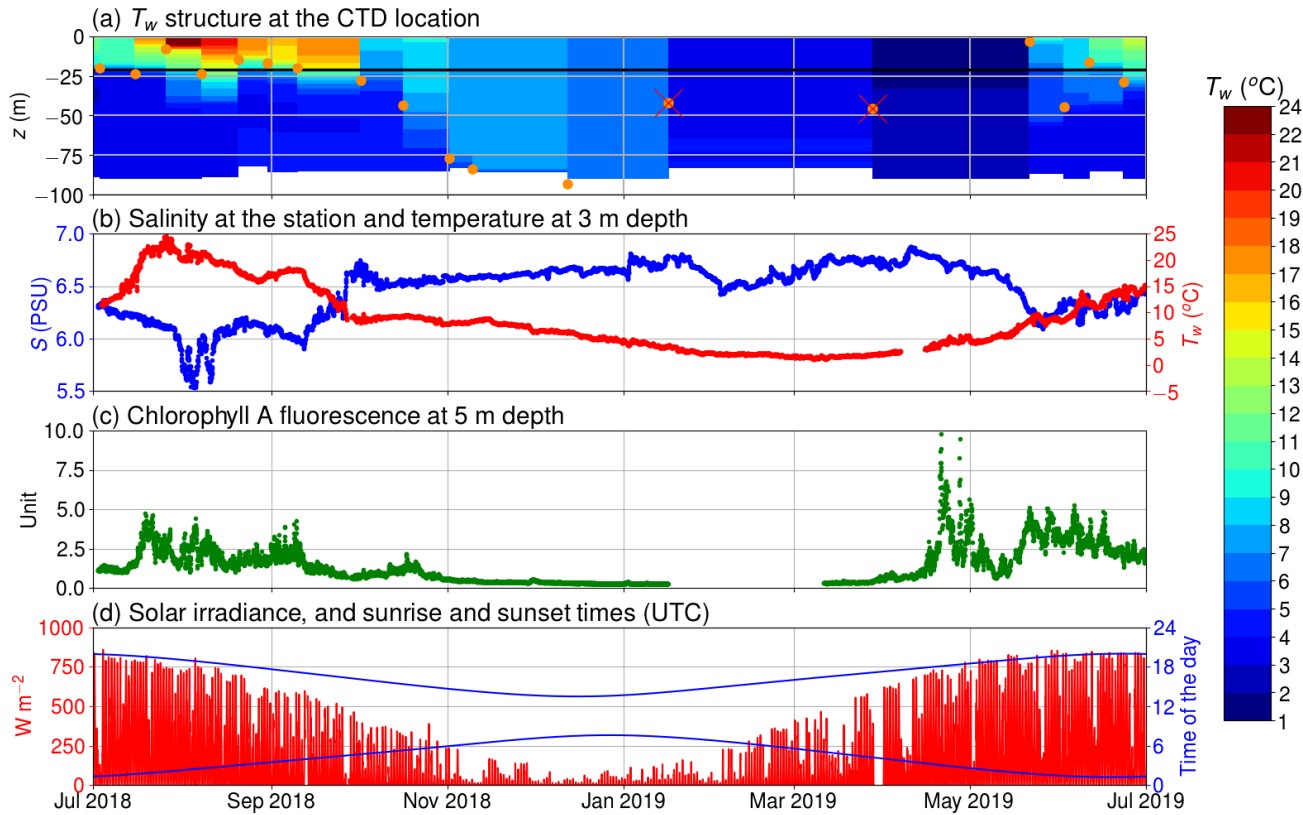

**Figure 2.** (a) The temperature of the seawater ($T_w$) assessed by the CTD casts and the depth of thermocline (orange circles); discarded thermocline depths are marked with a red cross, the horizontal black line depicts the depth of the inlet; (b) salinity at 5 m depth and temperature at 3 m depth; (c) chlorophyll A relative fluorescence at 5 m depth; (d) solar irradiance (in red) and sunrise and sunset times (in blue) in UTC.

Throughout the results, we use the range to describe the diurnal $pCO_2$ variability. The range, or the peak-to-peak amplitude, is defined as a difference between the diurnal $pCO_2$ maximum and minimum.

## 3 Results and discussion

### 3.1 Environmental conditions and seasonal $pCO_2$ variability

5   Our observations start in July 2018 during the so-called blue water period (Schneider and Müller, 2018), a phase in early summer that is characterised by close-to-zero net community production between the spring and the mid-summer bloom events (Andersson et al., 2017). As it is typical for this period, chlorophyll A concentration was low, which is reflected in a low relative fluorescence unit (Fig. 2c). At the same time, surface $pCO_2$ was close to equilibrium with the atmosphere. In mid-July, a cyanobacteria bloom developed, as it is typical for the study area and time of the year (Kraft et al., 2021). The primary

production activity lowered the $pCO_2$ below 200 µatm. This low $pCO_2$ level persisted for about one month (Fig. 3a). The measured oxygen concentration and calculated equilibrium concentration were close to equilibrium in the beginning of July, but due the cyanobacteria bloom, the oxygen concentrations diverged and for a week, the sea was strongly supersaturated. After the $pCO_2$ had increase to almost 600 µatm, another bloom occurred in early September and caused a second $pCO_2$ minimum.

After the another bloom, the measured oxygen stayed higher than the equilibrium concentration for a week. In late September 2018, $pCO_2$ peaked at 800 µatm. This is a result of the deepening of the mixed layer depth (Fig. 2a) which causes vertical entrainment of sub-thermocline water masses that are enriched in DIC due to the remineralization of organic matter. During winter time, the $pCO_2$ slowly decreased and reached equilibrium with the atmosphere by the end of March. Also, thorough the winter, the sea was mostly a sink of oxygen and the measured oxygen concentration predominantly increased. Chlorophyll

A fluorescence peaked again in April 2019 as a result of the spring bloom. Simultaneously, the $pCO_2$ dropped to 200 µatm, where it stayed for two months. The measured oxygen peaked at 475 µmol at the end of April, and the sea was supersaturated with oxygen for over two months.

Over the course of the year studied, the sea was a sink of atmospheric carbon for approximately four months. Generally, the seasonality of surface $pCO_2$ at Utö is similar to the open pelagic conditions in the Baltic Proper (Wesslander et al., 2010;

Schneider and Müller, 2018) but the maximum value (800 µatm) in autumn is considerably higher than observed in the Baltic Proper (600 µatm). This could be due to the fact that the water depth at the sampling location is low and thus remineralised $CO_2$ from the sediment surface can directly be entrained into surface waters upon vertical mixing.

The thermocline was located at the depth of 20 m during most of the time in summer 2018. In autumn, the thermocline deepened and in winter the water column was considered to be completely mixed. The thermocline may have only been

shallower than the inlet depth of the seawater supply occasionally, e.g., in spring 2019, when a shallow thermocline formed for a short period. Therefore, most of the time our flow-through setup was supplied with water from the mixed layer. We did not observe surface freshwater layers or permanent ice coverage during the measurement period that would be of relevance for the interpretation of our findings.

## 3.2 Examples of diurnal $pCO_2$ variability

Two contrasting examples of the diurnal $pCO_2$ variability at the beginning of September and in late December 2018 are shown in Fig. 4. On September 3, 2018, we observed a large diurnal $pCO_2$ range (maximum–minimum) of 108 µatm. The oxygen-derived biological $pCO_2$ signal shows a very similar pattern, indicating that this large $pCO_2$ diurnal variability is mainly a result of biological transformations. Minor deviations between observed $pCO_2$ and the biologically-driven changes based on oxygen dynamics occur early in the morning and late in the evening. The air–sea exchange had a negligible effect on the $pCO_2$

on that day, because the $pCO_2$ difference between the sea and atmosphere was close to zero. Including temperature as a driver into our model of the surface $pCO_2$ variability slightly increases the deviation from the observed hourly changes. It is possible that this is due to a too low oxygen-derived biological component. In Sect. 3.2.5, we give evidence of a slightly too small biological component in September.

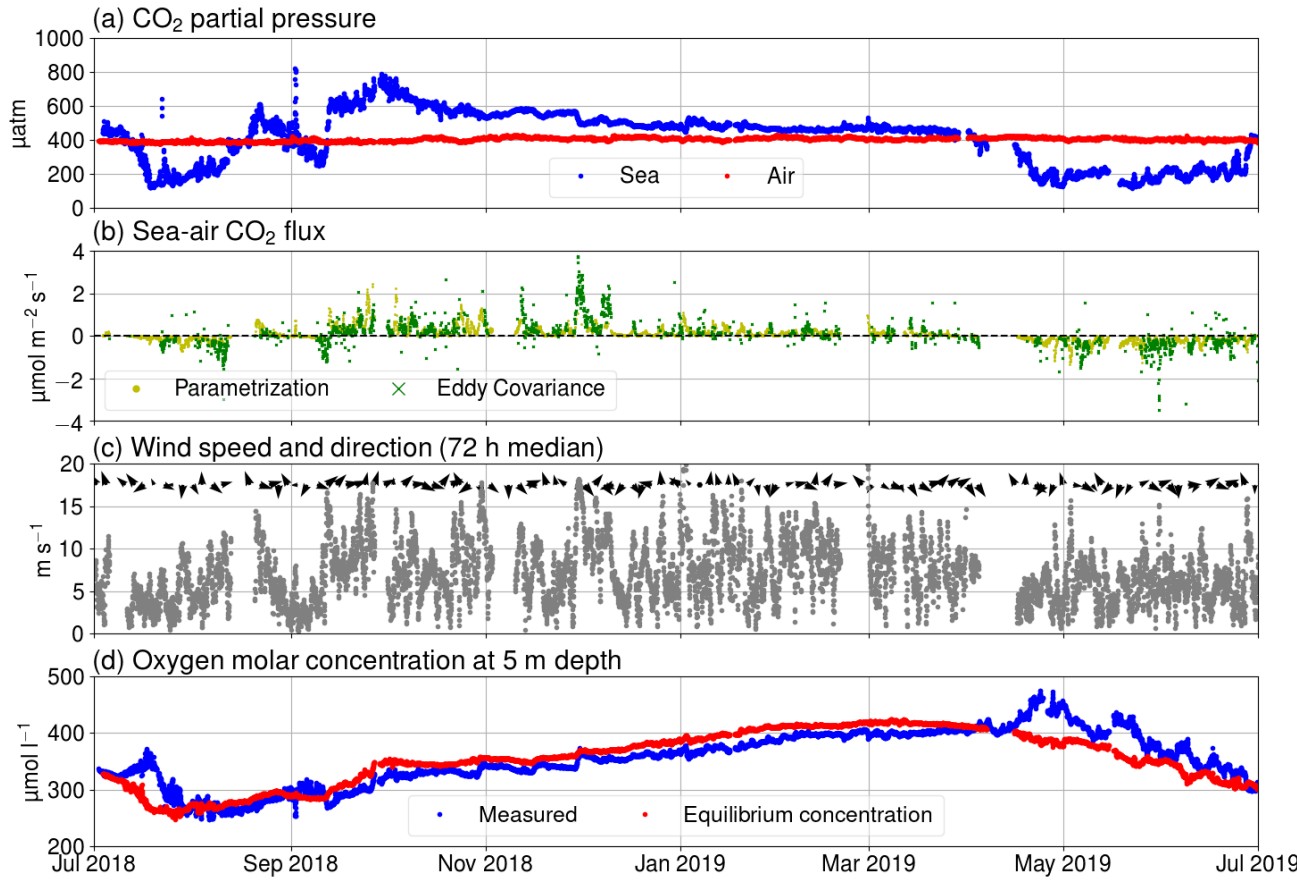

**Figure 3.** (a) $pCO_2$ in air (red) and seawater (blue), (b) $F_a s$ measured using the eddy covariance method (green) and calculated using Eq. A3 (yellow), (c) wind speed (gray dots) and direction (black arrows), and (d) oxygen molar concentration of seawater as measured (blue) and calculated for hypothetical equilibrium with the atmosphere (red).

The $pCO_2$ on December 20, 2018, was decreasing, almost linearly. This example shows that the oxygen-derived $pCO_2$ variation is higher than the observed $pCO_2$ variation in winter. The oxygen is primarily altered by mixing and air-sea exchange of oxygen. This issue is discussed in the chapter 3.2.5. Both the air-sea exchange of carbon and gradual cooling of the water contribute to the decrease of surface $pCO_2$.

The largest daily $pCO_2$ range (503 µatm) was detected on July 22. This extreme case can be attributed to an upwelling event as the water at the marine station, measured by the thermosalinograph, cooled by 5 °C simultaneously. Most of the cooling effect did not reach the thermistor at 3 m, as the temperature at the thermistor chain cooled less than 2 °C at 3 m depth. Observations made during this upwelling event were discarded from the following analysis of the diurnal $pCO_2$ variability. Another large $pCO_2$ change (452 µatm) occured on September 2, but the water temperature at the station changed approximately 1 °C, and thus we did not exclude the data from this day from our analysis.

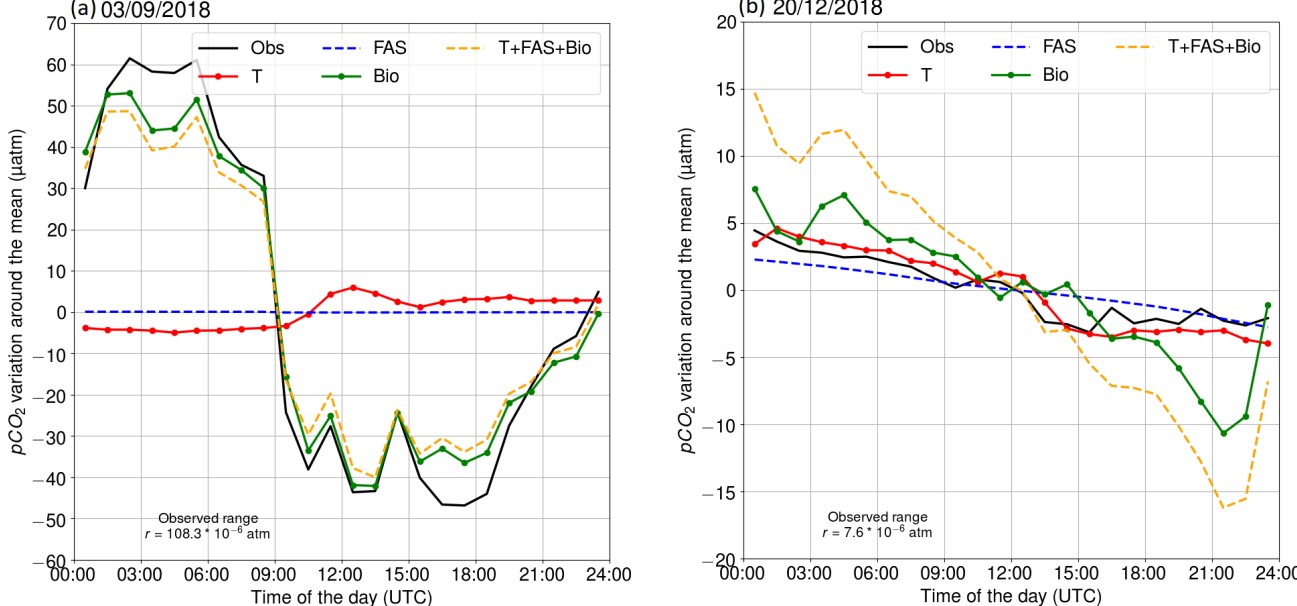

**Figure 4.** Diurnal $pCO_2$ variability on (a) September 3, 2018, and (b) December 20, 2018. The black line is the observed (Obs) evolution. Other lines represent the calculated $pCO_2$ evolution, driven by different processes: red indicates temperature (T), blue indicates the air–sea exchange of carbon dioxide (FAS), green indicates biological transformations (Bio), and orange indicates the combined effect of all processes (T + FAS + Bio).

### 3.2.1 Observed diurnal $pCO_2$ variability

The observed diurnal variability of $pCO_2$ was lowest during the winter time (Fig. 5). On average, the monthly median range (maximum – minimum) in November–February was only 4 µatm. Within the winter months, February revealed the lowest monthly median range and the lowest range between the 10th and 90th percentiles: less than 11 µatm daily variation were
5   observed for 80% of the time. In winter time, no clear diurnal pattern is visible, which goes along with varying times for the daily minimum and maximum $pCO_2$. This absence of a diurnal pattern in $pCO_2$ during winter is consistent with the findings of Lansø et al. (2017) for the Baltic Sea Proper.

In April, the observed diurnal $pCO_2$ variability starts to show a sinusoidal form, which remains until October. The diurnal $pCO_2$ minimum occurs during the afternoon and the maximum in early morning. At approximately 09:00 UTC (12:00 local
10   summer time), the $pCO_2$ is closest to the diurnal mean. The monthly median range of $pCO_2$ increased until August, which had the highest monthly median range of 31 µatm. In the Baltic Proper, the highest diurnal $pCO_2$ variability (27 µatm) was observed in September (Lansø et al., 2017). However, this difference is likely due to the interannual variability as different years are compared. There is large variability in diurnal $pCO_2$ over the course of a single month during the productive season. During

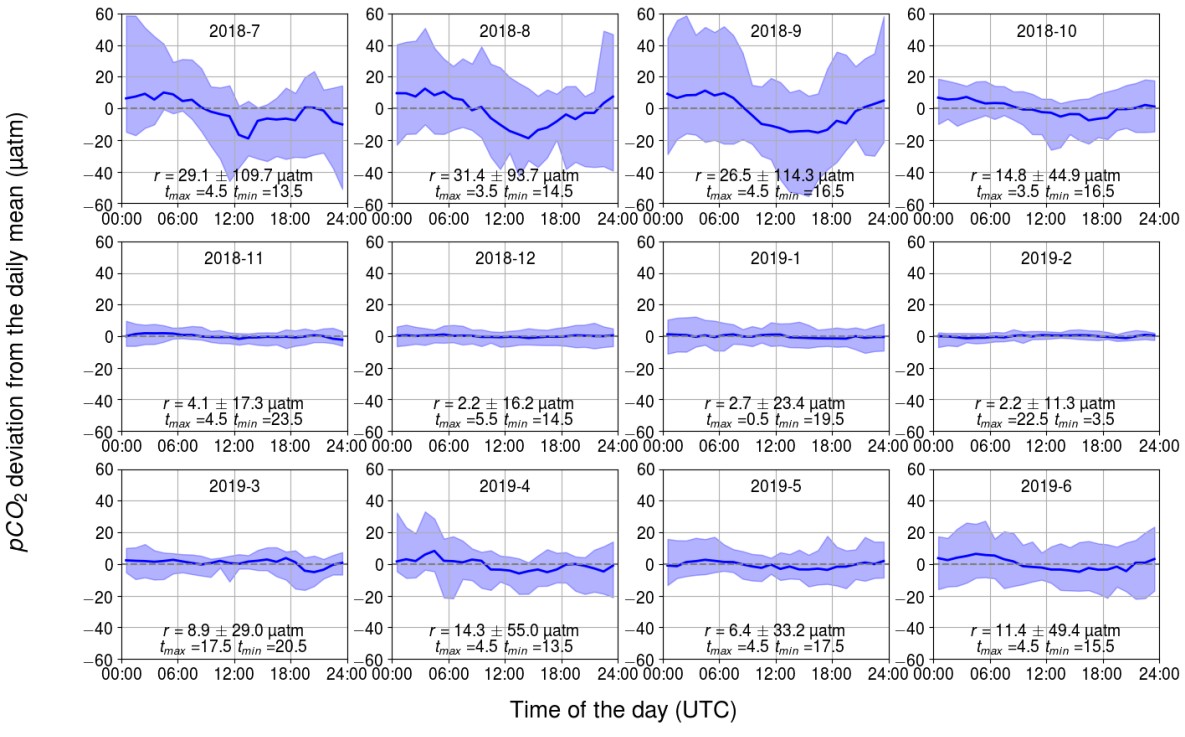

**Figure 5.** Observed diurnal $pCO_2$ variability. Displayed are the hourly binned median values (line) and the range between the minimum of the 10th percentile and the maximum of 90th percentile (ribbon). The $y$-axis shows the $pCO_2$ deviation in µatm and the $x$-axis shows the hour of the day. The mean and standard deviation of the daily range, $r$, and the timing for the maximum and minimum $pCO_2$ are also given.

this time, a single day may deviate significantly from the monthly median value. According to the 10th and 90th percentiles, 80% of the days in September occur within a large range of $114\,\mu\mathrm{atm}$.

### 3.2.2 Biology-related diurnal $pCO_2$ variability

The diurnal $pCO_2$ variability induced by biological activity and inferred from changes in the oxygen concentration, are closely
5   similar to the observed $pCO_2$ dynamics (see Figs. 4, 5, and 6). In both cases, sinusoidal diurnal variability with the maximum in the morning and the minimum in the afternoon during April-September is observed and the monthly median ranges are of similar strength. During nighttime, respiration (both heterotrophic and autotrophic) prevails, which increases DIC and thus also $pCO_2$. Solar irradiance intensifies as the day progresses and the carbon fixation outweighs the respiration, causing DIC to decrease. For our shallow sampling location, it is further possible that benthic processes impact surface water carbon dynamics,
10  especially when the water body is completely mixed.

In summer, the daytime increase in temperature partly counterbalances the $pCO_2$ reduction caused by primary production. The temperature-driven diurnal $pCO_2$ maximum and the biologically controlled $pCO_2$ minimum occur at approximately the same time in the afternoon. However, the temperature effect is significantly smaller than the impact of primary production.

The largest observed and modeled biological $pCO_2$ diurnal variability occurs in August and is twice as large as the range observed during the spring bloom. On the one hand, the temperature is at its annual maximum during July–August, which favors phytoplankton growth (Trombetta et al., 2019), but on the other hand, the solar irradiance is already decreasing from its annual maximum during June–July. During the spring bloom, chlorophyll A fluorescence was high compared with the one during August, when the highest $pCO_2$ variation is observed. However, microbial respiration tends to increase towards higher temperatures (Lopez-Urrutia et al., 2006), and thus the highest respiration rates are expected during July–August, contributing to the large amplitude of the diurnal cycle. It is possible that in spring, the daily $pCO_2$ range is lower than in autumn due to the deeper mixed layer in spring (Fig. 2a) causing the production to be distributed across a larger water volume.

Our data set suggests that, on average, the biological component controls $pCO_2$ diurnal variability, but on specific days during the biological season, other components (especially mixing) can have a stronger impact, as Wesslander et al. (2011) have shown.

During winter, the diurnal $pCO_2$ pattern generated by the biological processes revealed a positive trend over the course of a day, which could indicate the remineralization of organic matter. The fact that this directional trend is not seen in the observed $pCO_2$, could be due to the $CO_2$ release to the atmosphere counterbalancing the biological effect. However, it is implausible that the remineralization occurs for the whole winter and is even strongest in February.

### 3.2.3 Temperature-related diurnal $pCO_2$ variability

The daily variation in seawater temperature follows the cycle of solar irradiation. The highest monthly average of daily temperature range (daily maximum temperature – daily minimum temperature) was in July with 1.6 °C and the lowest in February with 0.2 °C.

The diurnal $pCO_2$ variability driven by changes in temperature is generally small (Fig. 7). Apart from June, July, and August, the monthly median range was 3 µatm or less. The largest monthly median range occured in July (5 µatm), when the solar irradiance reaches its annual maximum (Fig. 2e). Still, for 20% of the days in July, a temperature-related diurnal variability of $pCO_2 > 27$ µatm was observed.

During months with high solar radiation, i.e. March–September (Fig. 2e), the maximum of the temperature-related diurnal $pCO_2$ cycle occurs at noon and the minimum in the middle of the night or in the early morning. In winter, the temperature-related $pCO_2$ changes do not show a clear diurnal pattern nor directional trend.

The measurement depth of the temperature is 3 m. Directly at the sea surface, we would expect higher temperature-induced $pCO_2$ variability since solar irradiance decreases with depth.

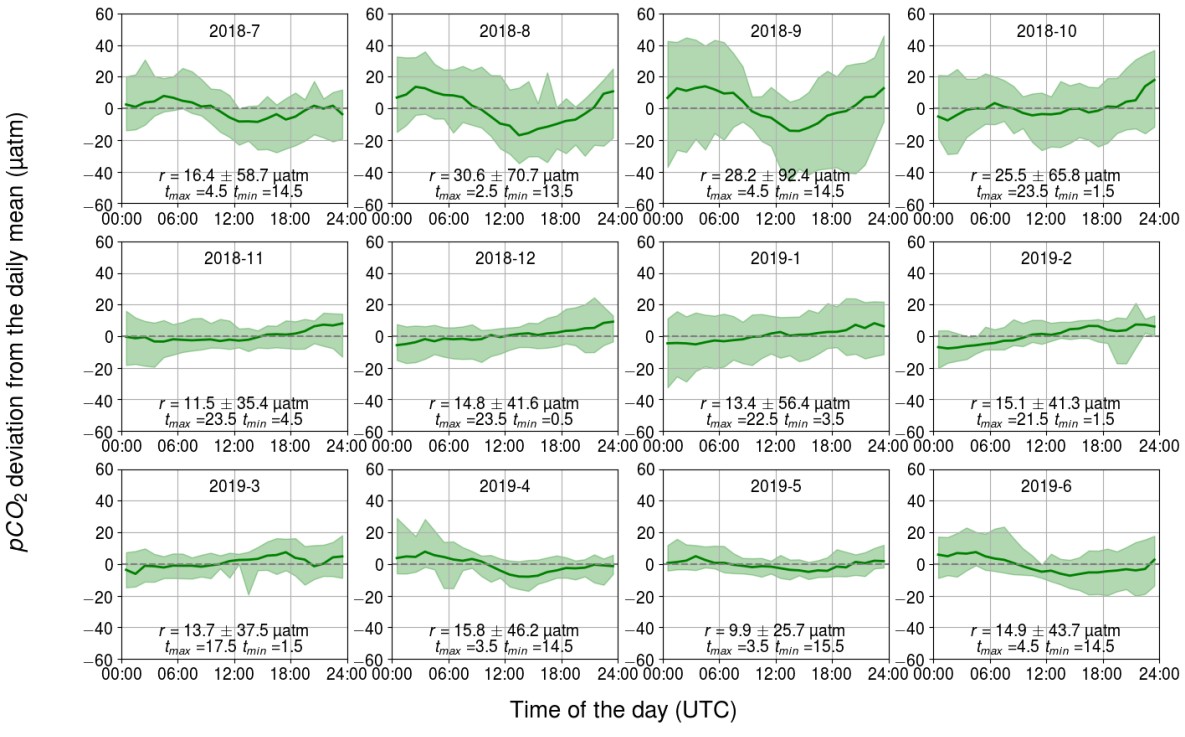

**Figure 6.** Observed monthly $pCO_2$ diurnal variability generated by biological transformations, showing the binned median and difference of the minimum of the 10th percentile and the maximum of the 90th percentile. The $y$-axis shows the $pCO_2$ deviation in $\mu$atm and the $x$-axis shows the hour of the day. Range, $r$, and the time for the maximum and minimum $pCO_2$ are also given.

### 3.2.4 Diurnal $pCO_2$ variability generated by the air–sea $CO_2$ flux

Diurnal $pCO_2$ fluctuations generated by the air–sea exchange of $CO_2$ exhibit a clear trend-like pattern (Fig. 8), due to the nature of the process. The direction of the air–sea $CO_2$ flux is controlled by the sign of the $CO_2$ partial pressure difference between the sea surface and the atmosphere. As the atmospheric $pCO_2$ is relatively stable compared to that of the sea, the flux direction is largely controlled by the seawater $pCO_2$. The trend in the diurnal pattern of $pCO_2$ generated by air-sea exchange thus represents the net carbon uptake of the Baltic Sea in summer when the sea surface $pCO_2$ is lower than atmospheric $pCO_2$ and vice versa in winter

The magnitude of the air–sea fluxes is largest during September–October when a large partial pressure gradient and high wind speeds co-occur. In these months, the monthly median range was $10\,\mu$atm or higher. In contrast, the effect of air–sea exchange on diurnal $pCO_2$ variability is almost negligible (less than $2\,\mu$atm) when the sea and atmosphere were nearly balanced with respect to $pCO_2$, as during December–March, or when the wind speeds are low, as in the summer months.

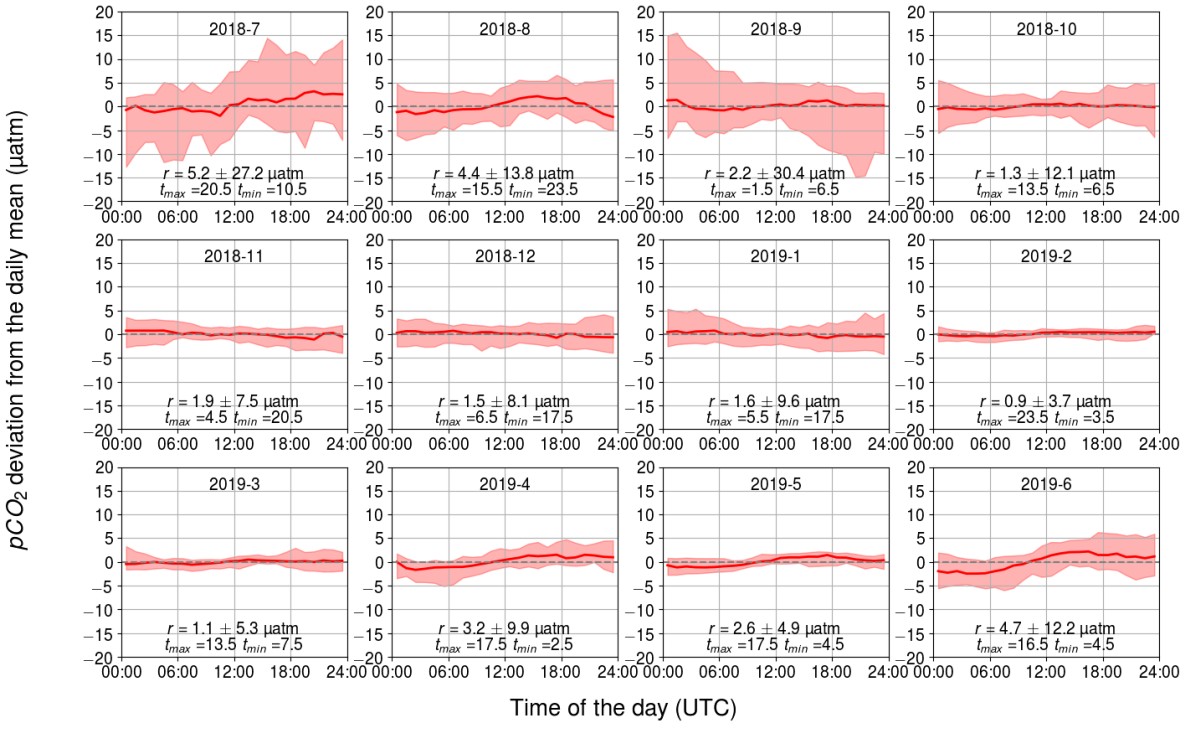

**Figure 7.** Temperature-induced cumulative daily changes in $pCO_2$, shown as the monthly climatological median and the difference of the minimum of the 10th percentile and the maximum of the 90th percentile. The $y$-axis shows the $pCO_2$ deviation in $\mu$atm and the $x$-axis shows the hour of the day. Range, $r$, and the time for the maximum and minimum $pCO_2$ are also given.

### 3.2.5 Comparing observed and estimated $pCO_2$ variability

When comparing the observed hourly change in $pCO_2$ and the calculated change that takes into account the three processes air–sea exchange, biology, and temperature (Fig. 9), we found that the overall $RMSE$ between all hourly modeled and observed $pCO_2$ changes was $10\,\mu$atm. $RMSE$ was $9$–$14\,\mu$atm during July–October, while it was less than $3$–$6\,\mu$atm during the other seasons. The scatter in Fig. 9 is visibly highest during July–October. These months showed the highest observed diurnal $pCO_2$ variability, which may have a direct effect on the increased error. For each month, we divided the $RMSE$ value with the average absolute change in hourly $pCO_2$ and found this ratio to be 1.26 on average during March–October, whereas during November–February it was 3.29 on average. Thus, the error introduced by the model during these winter months, though comparatively small in its absolute value, is large compared to the observed variability, which suggests that the estimates of the biological component during the winter time should be interpreted with care. This, however, does not have a significant effect on the analysis, since the biological activity in winter is negligible (see Fig. 2c).

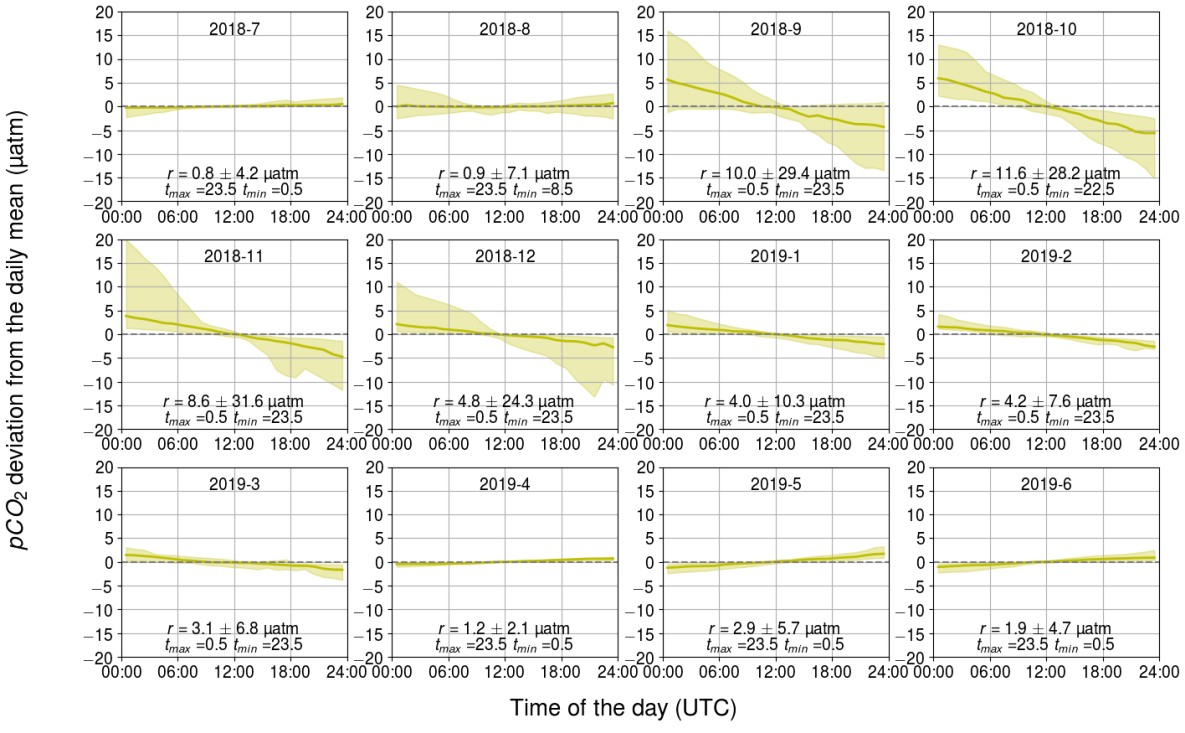

**Figure 8.** Monthly $pCO_2$ diurnal variability generated by the air–sea exchange of carbon dioxide, showing the binned median and difference of the minimum of the 10th percentile and the maximum of the 90th percentile. The $y$-axis shows the $pCO_2$ deviation in µatm and the $x$-axis shows the hour of the day. Range, $r$, and the time for the maximum and minimum $pCO_2$ are also given.

The fitted slope between the modeled and observed hourly $pCO_2$ changes appears to vary during the seasons. During the early winter months (November–January), the modeled $pCO_2$ changes are twice as large as the observations (see the slope of 2.1). During the late winter (February–March), the model and observations give the closest match with the slopes of 1.0–1.3. From April to October, the slope varied between 0.3 and 0.7, with the smallest slopes in July (0.3) and May (0.4).

5     Most of the variation in the modeled $pCO_2$ originates from the oxygen-derived biological processes, and thus we argue that the different slopes in observations and modeled data are related to the parameterization of the biological processes. To identify the reason for the mismatch between model and observations, we performed a similar analysis as in Fig. 9 but seperately disabled the oxygen flux between the atmosphere and sea (i.e. assuming all oxygen changes to originate from the biological transformations), , as well as temperature-induced $pCO_2$ changes and air–sea $CO_2$ flux, but these modifications of

10     our $pCO_2$ model proved to only have a negligible effect on the slopes. Possible remaining sources of error thus include the

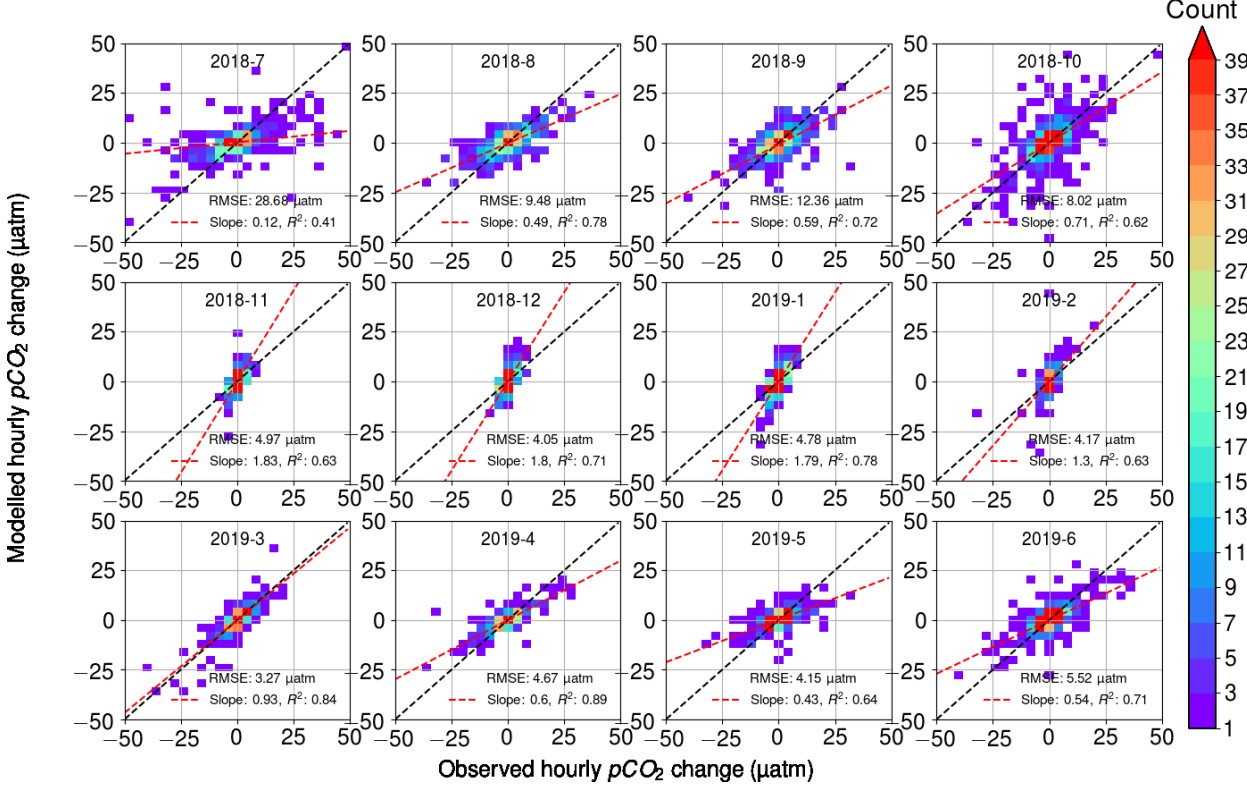

**Figure 9.** Modelled hourly $pCO_2$ changes as a function of observed $pCO_2$ changes. Color indicates the number of observations within bins of 2 µatm width. For each month, the $RMSE$ between the model and the observations is given, as well as the slope of the best fit (red line) with its correlation coefficient. The black line is the identity (1 : 1) line.

parameterization of the air-sea exchange of oxygen, the parameterization of the mixed layer depth and the carbon–oxygen ratio in Eq. 4.

It is possible that the seasonal slope changes in Fig. 9 are due to the fact that the oxygen concentration change are not well-constrained by the $O_2$ flux. This could be due to a time lag between the $O_2$ flux at the air-sea interface and the $O_2$ concentration change at 5 m depth. It is indeed likely that the wind speed-parameterization of $O_2$ flux provides a good estimate of the $O_2$ air-sea flux, but that the flux at the surface is challenging to translate into the $O_2$ concentration changes at 5 m depth at one hour resolution. In summer, the oxygen flux is directed from the sea to atmosphere, and thus its effect on the biological component during daytime should be positive. If this process is not taken into account, we might end up with an underestimated biological component, i.e. low slopes in Fig. 9. In winter, vice versa would happen.

A bias in our estimation of the mixed layer depth may also introduce an error in the modelled $pCO_2$ change. It is possible that in spring, the vertical redistribution of surface $O_2$ fluxes may not extend to the mixed layer depth. This would cause the gas exchange term of oxygen to be underestimated in Eq. 4, leading to the biological $pCO_2$ component in the model to be too low. In autumn, the calculated mixed layer depth might be too shallow to fully capture the vertical mixing of surface $O_2$ fluxes. A major limitation in this regard is our definition of the mixed layer depth as the water depth at the sampling location in cases when the true mixed layer depth at the CTD location was found deeper than the water depth in the inlet location. This limitation is critical, because it would not capture the loss of $O_2$ due to lateral mixing with deeper waters close to the sampling location. This would cause the gas exchange to be overestimated and the biological $pCO_2$ component to be too high.

The Redfield ratio for $CO_2$–$O_2$ (-0.77) used in this study is based on an oceanic average (Redfield et al., 1963). To explain the slopes between the model and the observations (-0.3 to -2.1) would require a $CO_2$–$O_2$ ratio of -0.37 in winter and as high as -2.5 in some summer months. The $CO_2$–$O_2$ ratio of respiration (the respiratory quotient) depends on the organic substrate in question, the degree of its oxidation, and the methabolic pathway used. This quotient may indeed vary between -0.13 and -4.00 (Robinson, 2019). In contrast, the required photosynthetic quotient of -2.5 in July appears very high compared with typical values (Laws, 1991). Wesslander et al. (2011) for example determined the $CO_2$–$O_2$ ratio in April 2006 in the Baltic Proper to be -1.0, with some diurnal variation. We thus conclude that the changes in respiratory and photosynthetic quotients alone cannot explain the seasonality in the slopes.

### 3.3  Effects on the air–sea exchange of $CO_2$

The diurnal $pCO_2$ variability can have a significant effect on the instantaneous air–sea $CO_2$ fluxes. The sign of the integrated daily air–sea $CO_2$ flux can even change when the $pCO_2$ at the sea surface and in the atmosphere are close to equilibrium, as was observed on the July 22 and on September 2 (data not shown).

The largest observed monthly median ranges in $pCO_2$ occurred during July–September (27–31 µatm). During this time the $pCO_2$ varied from slightly above 100 µatm to 800 µatm. In addition to the wind speed, the $pCO_2$ difference between the sea and the atmosphere controls the air–sea flux. The greatest relative effect on the daily flux occurs when the sea $pCO_2$ varies close to the atmospheric $pCO_2$, i.e., at approximately 400 µatm. In late July and early August 2018, the sea was a sink and in late August and September, the sea was a source of $CO_2$ to the atmosphere at the study site. The diurnal $pCO_2$ variability during these months are similar, with a maximum before noon and a minimum in the afternoon. However, in late July and early August, the $pCO_2$ difference between the sea and atmosphere is smallest before noon and largest in the afternoon, whereas in late August and September, the situation is reversed: the largest difference is before noon and the smallest is in the afternoon.

The discussion above only takes into account the diurnal variability of the air–sea $pCO_2$ gradient even though the flux also depends on the gas transfer velocity. This might also exhibit diurnal cyclicity, especially during clear skies in the coastal regions, where spatially uneven heating of the ground generates pressure gradients and thus winds. The most popular parameterizations for gas transfer velocity are either quadratic or cubic functions of the wind speed and thus even small changes in wind speed have large impact on the flux.

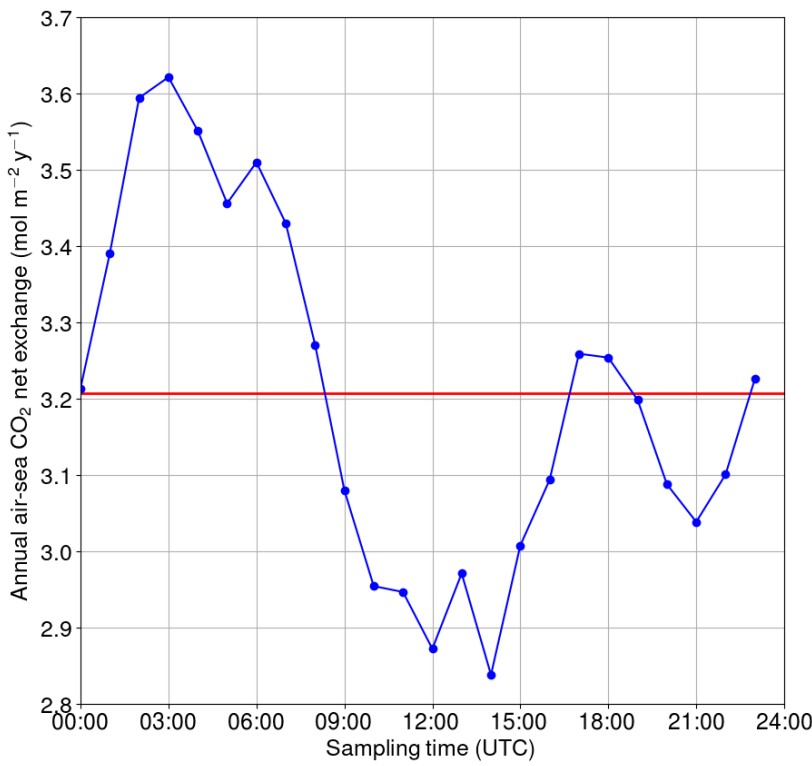

**Figure 10.** Annual net exchange of carbon dioxide between the sea and atmosphere if only one measurement per day is used. The reference (the red line) is based on high-frequency data.

For the hypothetical case of a single sampling event per day, we calculated how the annual net exchange of carbon dioxide between the sea and atmosphere would vary depending on the sampling time (Fig. 10). The calculations were performed using the flux parameterization of Wanninkhof (2014). The reference net exchange (red line in Fig. 10, i.e. the "true" value) is calculated using the high-frequency one-hourly data, whereas the other fluxes are calculated using only one measurement at the daytime indicated on the x-axis. The closest match with the "true" net flux is achieved when sampling the seawater at 09:00, 17:00-18:00 or 24:00 UTC. In contrast, sampling between 00:00 and 09:00 UTC causes an overestimation of the net flux by up to 12%, whereas sampling between 09:00 and 18:00 UTC leads to an underestimation of up to -12%. The sinusoidal shape of the net flux bias as a function of the sampling time clearly originates from the biological component of surface $pCO_2$, but the deviation from the sinusoid around 15:00–20:00 UTC must originate from the turbulence parameterization (wind speed) as such a shape is not observed in the $pCO_2$.

## 4 Conclusions

The diurnal variability of sea surface $pCO_2$ and the contributions of its drivers were studied at Utö station in the Archipelago Sea of the Baltic Sea. Multiple processes affecting the diurnal $pCO_2$ variability at Utö were distinguished and their interplay was found to depended on season, similarly as previously shown for the East of Gotland by Wesslander et al. (2011). At Utö, the largest variability was found during July–September, when the monthly median of the diurnal $pCO_2$ varied in the range of 27–31 $\mu$atm. This $pCO_2$ variability was mostly generated by the biological transformations (i.e. the production and respiration or organic matter). However, individual days showed significantly higher variations. Extreme $pCO_2$ variations exceeded 500 $\mu$atm a day and were attributed to upwelling of $CO_2$-enriched water masses. Diurnal $pCO_2$ variability was less pronounced in winter time, which is comparable to the observations in the Baltic Proper (Lansø et al., 2017). Thus, on average, the magnitude and the timing of the diurnal $pCO_2$ variability at Utö are similar to the ones of the pelagic conditions in the Baltic Proper, except for coastal upwelling at the study site.

Assessment of the annual air–sea flux based on the entire data set or individual one-hour sampling times revealed a potential bias caused by the time of sampling of up to 12%. This finding suggests that data from moving platforms which do not resolve the diurnal cycle, like research vessels or VOS lines, can lead to substantial biases in flux calculations or the estimation of natural variability.

These findings emphasize the importance of continuous measurements at fixed locations providing a high temporal resolution, in order to complement VOS-based observations that achieve high spatial coverage. Our autonomous high-frequency measurements of the seawater carbonate system at fixed sites has proven to be valuable in the assessment of the short-term variability of the carbonate system. However, as European seas are spatially highly heterogeneous, our findings call for organized efforts to map the diurnal variability of the carbon system.

*Data availability.* TEXT

The data used in this paper can be found in the Zenodo repository (https://doi.org/10.5281/zenodo.4292384).

## Appendix A: The air–sea exchange of $CO_2$

The $CO_2$ exchange between the atmosphere and the sea, $F_{as}$, is driven by the difference in $CO_2$ partial pressure ($\Delta pCO_2 = pCO_2 - pCO_2^{atm}$) between the surface seawater and atmosphere, or more precisely, the differences in fugacity, which refers to the effective partial pressure of $CO_2$ that takes into account the non-ideal gas behaviour of $CO_2$. $CO_2$ partial pressure and fugacity only differ slightly and, for this reason, only partial pressure is used from now on. The efficiency of the exchange through the diffusive boundary layers of the gas and liquid fluids is defined by the gas transfer velocity, $k$. Thus, $F_{as}$ may be written as:

$$F_{as} = kK_0\Delta pCO_2, \tag{A1}$$

where $K_0$ is the solubility of $CO_2$.

The effect of the kinematic viscosity of seawater and the diffusion efficiency of $CO_2$ on $k$ are taken into account by including the ratio of momentum diffusivity in mass diffusivity, the Schmidt number ($Sc$), in $k$:

$$k = k_{660} \left( \frac{Sc}{660} \right)^{-1/2}.$$ (A2)

Since the Schmidt number is a function of temperature, it is normalized with the $Sc$ of seawater at $20\,°C$, a value of 660. A wind speed measured at $10\,m$ ($U_{10}$) is most commonly used to parameterize $k_{660}$, and probably the most well-known parameterization is a quadratic relationship proposed by Wanninkhof (1992), which was revised by Wanninkhof (2014):

$$k_{660} = 0.251 U_{10}{}^2.$$ (A3)

## A1   The parameterization of gas transfer velocity

We patched the $CO_2$ air–sea flux time series using the $U_{10}$ based parameterization for $k_{660}$ proposed by Wanninkhof (2014). The applicability of this parameterization for the western marine region of Utö was assessed by calculating the absolute value of $k_{660}$ from the measured $CO_2$ air–sea flux (from eddy covariance), partial pressure difference, solubility (Weiss, 1974), and the Schmidt number (Wanninkhof, 1992). Only cases with southwestern (180–330 $°$) winds and strong $pCO_2$ difference (>30 $\mu atm$) were considered. $CO_2$ flux outliers were discarded so that we only included the fluxes that are within two standard deviations from the median.

Non-stationarity is one of the determinant factors for the quality of direct flux measurement, and thus, non-stationary fluxes are discarded. Here, this means that the mean of $5\,min$ fluxes can deviate less than 30% from the $30\,min$ flux. The fully stationary condition is purely a theoretical concept, and the threshold for the accepted deviation from this is a matter of choice.

The best quadratic fit ($0.37\,U_{10}^2$) is somewhat larger than the parameterization proposed by Wanninkhof (2014), which might indicate enhanced gas transfer due to the coastal characteristics of the study site. However, for the comparability, we stick with the common parameterization by Wanninkhof (2014). Low and medium wind speeds are well packed, whereas the 10th and 90th percentiles move further away from each other at high wind speeds. The parameterization of Wanninkhof (2014) shows the highest deviation from the binned median values at highest wind speeds. The binned median at the highest wind speeds is low compared with the results of Wanninkhof (2014), which may indicate fetch limitation. More observations at high wind speeds is thus required for the in-depth analysis.

## Appendix B:  The inorganic carbon system

Gaseous $CO_2$ dissolves into water, where part of it hydrates into carbonic acid ($H_2CO_3$). Dissolved $CO_2$ and carbonic acid are not easily distinguished, and thus the sum of their concentrations is denoted as $[CO_2^*]$:

$$[CO_2^*] = [CO_2] + [H_2CO_3].$$ (B1)

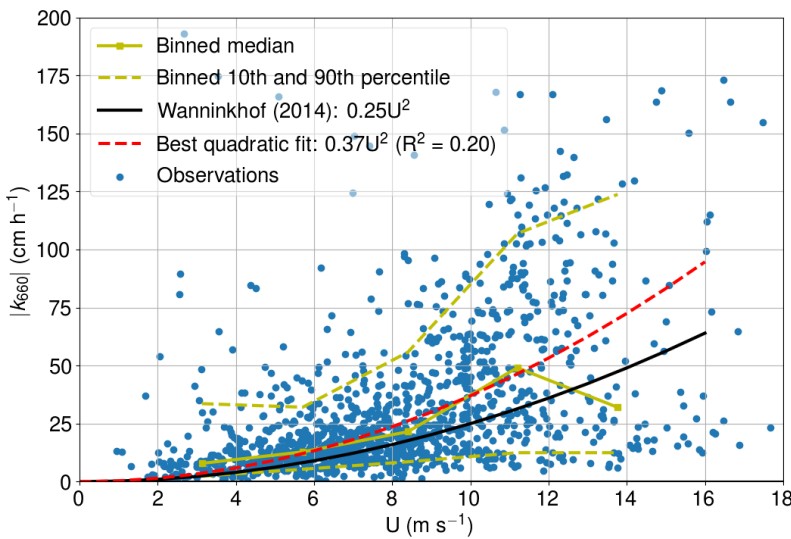

**Figure A1.** Measured gas transfer velocity as a function of wind speed.

Henry's law describes the relationship between the fugacity of gaseous $CO_2$, which is in equilibrium with the underlying water, and the dissolved concentration of $CO_2$:

$$K_0 = [CO_2^*]/pCO_2. \tag{B2}$$

Carbonic acid dissociates to hydrogen carbonate ($HCO_3^-$, also known as bicarbonate), which further dissociates to carbonate
($CO_3^{2-}$) and hydrogen ions. The equilibrium states:

$$K_1 = \frac{[H^+][HCO_3^-]}{[CO_2^*]}, \tag{B3}$$

$$K_2 = \frac{[H^+][CO_3^{2-}]}{[HCO_3^-]}. \tag{B4}$$

Solubility and dissociation constants ($K_1$ and $K_2$) depend on the free energy of the reaction and thus are functions of temperature and pressure. As these stoichiometric constants are defined using concentrations instead of ion activities, they are also a
10 function of salinity.

Dissolved carbon dioxide, carbonic acid, bicarbonate and carbonate ions form the pool of total dissolved inorganic carbon (DIC):

$$DIC = [CO_2^*] + [HCO_3^-] + [CO_3^{2-}]. \tag{B5}$$

DIC is a conservative quantity, i.e., it does not vary as temperature or pressure change. The concentrations of different DIC
species change, but the sum of these concentrations remains the same if no carbon is added to or removed from the system.

If nutrients and photosynthetically active radiation are available, dissolved $CO_2$ is transformed into organic matter through the process of photosynthesis. When phytoplankton and other aquatic organisms respire, the opposite occurs and $CO_2$ is released. Through microbial degradation in water or in sediments, dissolved organic matter is transformed again into inorganic carbon.

Of all the parameters of the carbonate system, one can only measure $pCO_2$, DIC, TA, and pH (the negative logarithm of hydrogen concentration). To gain the complete description of the carbonate system, one should know at least two of these variables in addition to the information on seawater temperature ($T$), salinity ($S$), and pressure ($P$). Ideally, the effect of dissolved organic matter on total alkalinity should also be known. From Henry's law (Eq. B2), we see that $CO_2$ fugacity depends on the solubility and dissolved $CO_2$ concentration. Both of these variables are functions of temperature, salinity, and

pressure. The non-conservativity of $[CO_2^*]$ is due to the effect of the dissociation constants, $K_1$ and $K_2$.

## Appendix C:  Total alkalinity

Another important variable for the carbonate system is total alkalinity (TA), which is defined as the excess of proton acceptors (acids) over donors (bases). For most practical purposes, it is sufficient to only include carbonate alkalinity, boron alkalinity, and a component from the self-dissociation of water (which is commonly referred to as practical alkalinity):

$$TA = \underbrace{[HCO_3^-] + 2[CO_3^{2-}]}_{\text{Carbonate alkalinity}}$$

$$+ \underbrace{[B(OH)_4^-]}_{\text{Borate alkalinity}}$$

$$+ \underbrace{[OH^-] - [H^+]}_{\text{Self-dissociation of water component}}$$

$$\pm \text{ minor TA components.} \tag{C1}$$

Minor TA components include organic ions, which may have a large regional impact. In the case of the Baltic Sea, the bulk

of dissolved organic matter has been shown to act as a proton acceptor (Kuliński et al., 2014). Similarly to DIC, TA is a conservative quantity.

    Calcium carbonate ($CaCO_3$) is formed in a slow precipitation process by specific calcifying organisms. The precipitation and dissolution of $CaCO_3$ affect both DIC and TA. However, in the case of the Baltic Sea, calcifying phytoplankton only exists in the areas next to the North Sea (Tyrrell et al., 2008), and thus, the formation of $CaCO_3$ can be excluded in calculations for

most parts of the pelagic Baltic Sea, including our study site. On the other hand, the weathering of fluvial $CaCO_3$ has a determinant effect on TA in the limestone-rich southern regions of the Baltic Sea (Müller et al., 2016).

    We used the pair of the $pCO_2$ and the TA in our carbonate system calculations. The TA is parameterized using the salinity, because both of these variables are affected by the conservative mixing.

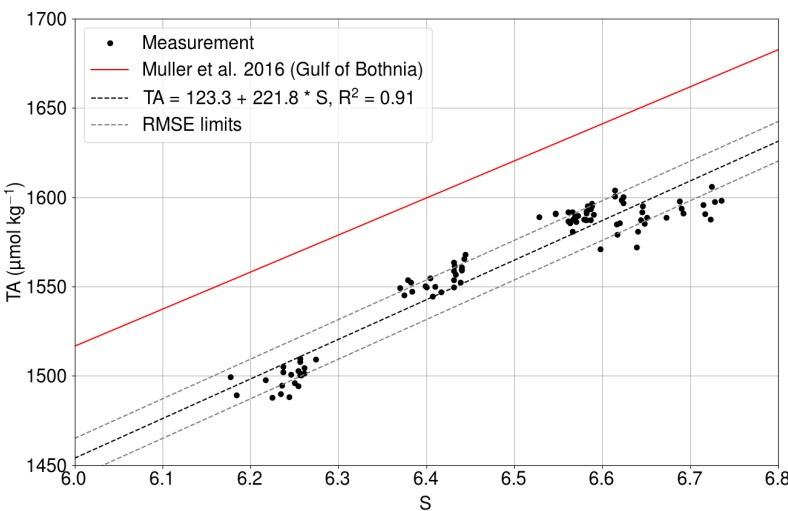

**Figure C1.** Measured total alkalinity (black dots) as a function of salinity at Utö in 2017 (Lehto, 2019). The solid red line shows the TA–$S$ relationship for the Gulf of Bothnia given by Müller et al. (2016), extapolated for 2018. The black dashed line is the best fit, and gray dashed lines show the same line with the limits of $RMSE$s.

The least squares fit of the relationship between the salinity and the directly measured total alkalinity (Fig. C1) had an $R^2$ value of 0.91. The $RMSE$ between the measurements and the fit is 11.1 $\mu$mol kg$^{-1}$. The slope is almost identical to the dependence found for the Gulf of Bothnia by Müller et al. (2016), extrapolated for the year 2018.

### C1  Processes controlling $p\mathrm{CO_2}$ omitted in the analysis

In our analysis to distinguish the different processes that drive $p\mathrm{CO_2}$ variability, we considered temperature changes, air-sea exchange of carbon and biological transformations. Several processes were omitted.

The salinity changes are related to mixing, and thus the interpretation of the salinity effect is not straight-forward and is not dealt with in this paper. The salinity effect on $p\mathrm{CO_2}$ is generally small: in oceanic conditions, a salinity change of 1 would generate a 9 $\mu$atm change in $p\mathrm{CO_2}$ (Sarmiento and Gruber, 2004). At Utö, the salinity varies less than 1.5 units during the

whole year (see Fig. 2). We neglect the effect of pressure on $p\mathrm{CO_2}$, because we interpret surface water $p\mathrm{CO_2}$ at one depth.

Some of these unknown drivers, such as mixing processes and freshwater effects, are assumed to be temporally random in nature, and thus their effect on $p\mathrm{CO_2}$ is considered to be negligible when inspecting average diurnal cycles. Some of the processes, e.g., alkalinity-related variations affecting $p\mathrm{CO_2}$, are unknown and may involve diurnal cyclicity. A salinity–alkalinity relationship used in the analysis takes into account the conservative variation of these variables due to the mixing and fresh-

water input. Nitrogen transformations during primary production can have a small effect on alkalinity that is not considered in the salinity–alkalinity relationship.

In general, the tidal force is the most prominent process to generate a diurnal pattern on the mixing of the DIC. In this location of the Baltic Sea, the effect of the tidal currents on the water masses is very small and thus can be neglected. However, several other processes such as the upwelling can also generate mixing. The driving force of the upwelling (or downwelling) is steady wind over the sea, and at our study site, open sea which contains very small islands, sea-breeze cannot be completely neglected but is not expected to be strong. However, there is a possibility that the density driven mixing has a diurnal cycle due to the diurnal heating/cooling of the surface waters.

The mixing component of the diurnal DIC variations can be large occasionally. For instance, there was clear indications of the mixing of water masses on July 22, 2018; the $p\mathrm{CO}_2$ varied by 503 $\mathrm{\mu atm}$ while the water cooled by 8 °C. However, there is not always that clear indicators suggesting the mixing events. In order to analyze the effect of the mixing on DIC precisely, one would need to know the 3D field of DIC and the water currents. This would require an array of carbonate system measurements. The analysis of the mixing of DIC is thus beyond the scope of this paper.

In the result and discussion section, we analyze the importance of individual drivers and the applicability of the method by comparing the calculated $p\mathrm{CO}_2$ changes to the observations.

*Author contributions.* TEXT

MH, LL, JS, JDM, and GR were in charge of the conceptualization. MH performed data analysis and visualisation. The manuscript was written by MH, LL, JS, JDM, and GR. SK, PY, LL, JS, and TM designed and constructed the flow-through system. TM and LL designed and constructed the flux setup. JH was in charge of data management.

*Competing interests.* TEXT

The authors declare that they have no conflict of interest.

*Disclaimer.* TEXT

*Acknowledgements.* Part of this work was supported by the JERICO-NEXT and JERICO-S3 projects, receiving funding from the European Union's Horizon 2020 research and innovation programme under grant agreements no. 654410 and no. 871153, respectively. The BONUS INTEGRAL project funded by the European Union (EU) and the Finnish Academy project SEASINK are also acknowledged for the partial funding of this research. The BONUS INTEGRAL project receives funding from BONUS (Art 185), funded jointly by the EU, the German Federal Ministry of Education and Research, the Swedish Research Council Formas, the Academy of Finland, the Polish National Centre for Research and Development, and the Estonian Research Council.

Finnish Marine Research Infrastructure (FINMARI) is acknowledged for the funding of the marine research instrumentation. We thank Ismo and Brita Willström for the CTD castings and maintaining the stations, and we thank Anne-Mari Lehto for providing the total alkalinity measurements. Also, thanks are due to the Integrated Carbon Observation System (ICOS) for providing the atmospheric $CO_2$ data at Utö. We acknowledge Theo Kurten for giving guidance in chemistry and Jani Särkkä for giving guidance in mathematical formulations. The CO2SYS program is acknowledged.

We thank the referees for their insightful comments resulting in highly improved manuscript.

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
