# Peer review of "The diurnal cycle of the $pCO_2$ in the coastal region of the Baltic Sea"

_Ocean Science, 2020_

## Referee Comment (RC1) · Anonymous Referee #1 · 22 Jan 2021

General comments

The paper presents and discuss one year of continuous measurements of marine CO2 concentrations, gathered at the marine monitoring station of Utö island, in the northern area of the Baltic sea. Overall this article addresses a timely topic and the results presented can substantially contribute to improving estimates of the CO2 emitted and absorbed by the Baltic Sea.

Marine data are mainly gathered by a flow through system that pumps seawater from 5 m depth and 250 m from the coast, allowing continuous measurements of the basic physical parameters as well as dissolved oxygen, fluorescence and CO2 partial pressure (pCO2) also during harsh wintry conditions. The marine dataset is completed with direct measurements of air-sea CO2 fluxes gathered by a flux tower on the seashore

and high quality atmospheric CO2 measurements acquired at the nearby meteorological station, making this dataset new and extremely valuable.

The data are analyzed to highlight the seasonal variability, the diurnal pCO2 cycle and to identify its main drivers. Finally, the authors estimate the bias that can be introduced calculating annual budget of the Baltic CO2 air-sea fluxes without taking into account this daily variability.

The main point where the paper needs a substantial improve is the overall writing and presentation of the methods and the results. I also recommend a revision of English style.

Methods used look generally adequate but are poorly written: you have to go back and forth the pages to understand what has been measured and where, and they often lack of detailed information on accuracy both on measured and calculated parameters. The model proposed to separate the different drivers of CO2 variability needs a better discussion mentioning also the variability that can derive from the advection of different water masses. Some trends presented in figure 6 and 7 are unexpected and need a better discussion.

The conclusions can be improved adding a comparison with other coastal areas / previous studies, where similar data are available.

Specific comments

Please, revise the title: the paper discus the daily variability of pCO2, and related air-sea CO2 fluxes, not of other parameters of the CO2 system as carbonate saturation state, DIC, pH...

Section 2 "Controls on the partial pressure of CO2"

This section is rather confusing. It summarizes some basic aspects of carbon chemistry and, at the same time, explains which can be neglected in the study area. I would suggest to present here the basic of carbon chemistry, with clear references to the ap-
pendix, and move the considerations on which process is important for the area, after the presentation of the study site, where the calculation performed are explained.

3 Materials and methods

Please, add a map with the location of Utö island, the station and the different sampling locations mentioned in the paper. The reference (Laasko et al 2018) includes mainly a description of the atmospheric station. This paper could be a good opportunity to present the Utö marine station with a table summarizing instrument type, variable measured, frequency etc.

The description of the instrumental setup and the whole method section is quite confusing and needs a reorganization where the description of measured data is clearly separated from calculated parameters and experimental considerations. I would suggest the following:

Improve the description of the study site providing general physical and biogeochemical settings of the study area. Describe the different stations where data are gathered and sampling strategy of all the data presented and discussed in the paper.

* Measurements from the flow-through system: instrument used, parameters acquired, frequency, accuracy, etc.

* Measurements from other stations: manual samplings, laboratory analysis, method, accuracy.

* Assessment of data quality etc.. you could add a separate section or add a sentence where there is the description of data acquisition.

* Calculated data: describe how you calculated all the derived parameters and estimate the uncertainty on derived carbonate system variables.

P8 The description of the calculation done is rather hard to follow. The DIC derived from mixing with other water masses is not considered in your model. Why? If you

consider that this is a minor component of DIC variability in the study area, you should explain and justify your choice.

4 Results and discussion

Section 4.2: why did you choose two days one after the other as an example? You could present two days from two different seasons or where the weight of the diverse pCO2 components is different. Otherwise you can show only one day of pCO2 data and add a picture with the daily variability of the other variables (temperature, oxygen,...).

P13: Add a description of the daily variability of temperature to improve the discussion and to allow a better comparison of your results with other areas

Both in figure 6 and 7 the data reported at h 00:00 are often markedly different from that reported at h24:00: they should be the same number or a very close one. Could you please explain better the data reported in your figures or double check your calculation?

Figures

Figure 1: It contains a lot of information but I found it hard to read. Please, increase the size along the "y" axis. Fig 1 (a): it's hard to see the red crosses and the black line. Is this temperature from monthly CTD casts in the nearby station? You should mention it in the caption. You could also present just the top of the CTD casts, down to the depth of the seafloor at the inlet position, and highlight the depth of the inlet to the flow-through system. Please add the conversion to local time in the caption as well.

Figure 2: Please, increase the size of the figure and of the symbols in the captions within the figures, especially that of figure 2 (b).

Appendix D

The gas transfer velocity for seawater normalized at 20°C (k660) is always a positive value. The direction of the flux is given by air-sea pCO2 difference: it is directed from the sea to the atmosphere (positive) if pCO2 sea> pCO2 atm and vice-versa. When

k660 is estimated from direct FCO2 measurements gathered by the eddy covariance method, the absolute value should be used. Making this correction to figure D1 the agreement between the experimental results and the theoretical model will greatly improve.

Technical corrections

Methods:

P.5 L3.: which instruments log data every 15 sec? The ones mentioned at the end of the page?

P5 L12: Is the gas calibration traceable to international CO2 standards (e.g. WMO, NOAA, ..)? If yes, please, add this information.

P5 L15-27: There is a mix of "calculated values" with measured ones, please, describe here only measured ones. Improve the description of temperature and salinity measurements: frequency, accuracy, frequency of calibration, etc. Describe the thermistor chain when you present the marine station. Add briefly the methods for TA measurements.

P5L 28: Even if in Kilkki et al., 2015 there is a detailed description of the methods, please, add the most important details: instrument model, accuracy, sampling frequency, reference to international standards.

P5 L30-34: Please, add detail on instrument accuracy, frequency of calibration etc..

P6 L1-12: it is not clear which data are reported in Figure 1.

P6 L8-22: move to the section of calculated parameters

P6 L22: delete "be"

P6 L23 Please, define "Fas" when it is used for the first time.

I suggest to delete the division in two subsections or modify the titles. As they are it

seems that "the flux parametrization" is related to the eddy covariance method, not a different method to calculate fluxes.

P7 L20 – 28: first describe how TA is calculated from salinity and the robustness of the choice done, then describe the calculations of the derived carbon chemistry parameters.

P7 L29 – P8 L3: Please, rephrase and explain better.

P8 L4-6: Please, describe step by step, it is quite confusing

P8 L22: Please rephrase, it is not clear what do you mean with ". . . each day at a time".

P8 L22-26: it is not clear, what pCO'2,i is

P8 L28-30: please, explain better, this point is not clear.

Results and discussion

P9 L5: A more appropriate title would be "Environmental conditions and seasonal pCO2 variability". Conclusions

P18 L30-32: All the parametrizations for gas transfer velocity contain a cubic o quadratic component function.

---

## Referee Comment (RC2) · Anonymous Referee #2 · 25 Jan 2021

Diurnal cycle of the CO2 system in the coastal region of the Baltic Sea, by Honkanen et al.

General comments

The paper by Honkanen et al. presents measured pCO2 data over a year from July 2018 to June 2019 from a marine site off the island Utö in the Baltic Sea. A variety of additional data is provided from the marine site, from two additional coastal sites and from the flux tower at the Utö island. The authors have examined the different drivers for the observed changes in pCO2, and they address the importance of high frequency data when precise air-sea CO2 fluxes are to be calculated. Data from coastal sites like this is highly valuable and the manuscript is very welcomed. However, the manuscript is poorly written and needs improvement both regarding English language and the

structure. In the current version, the reader needs to jump back and forth to get a grip on the story, and it is not always clear what the authors want to communicate.

Specific comments

Abstract: phrases like "monthly median diurnal pCO2 peak-to-peak amplitude" is difficult to understand. Please change to e.g. "monthly median of diurnal ..." throughout the manuscript

Chapter 1: The dataset are from 2018-2019, why not use a more updated reference for the global carbo cycle (e.g. the years 2009-2018: Friedlingstein et al., 2019, Earth Syst. Sci. Data, 11, 1783–1838, 2019, https://doi.org/10.5194/essd-11-1783-2019).

Chapter 2: Please note that Appendix A is not mentioned in the text.

Chapter 3 Materials and methods: This chapter needs improvement. It would be helpful if a map was included where the position of the Utö station, its water intake, thermistor chain position as well as the position of the ctd casts were marked. You should clarify the difference between the marine station and station, maybe it is better to use "laboratory" for the on-shore facility? LI-COR 840 measures CO2 and H2O, but are the gas dried prior to the detection (if so, please include information). Are you using the H2O measurements to calculate dry xCO2, or are you calculating pH2O from T and S? The pCO2 is corrected for temperature difference between water intake and equilibrator, and this should be done according to Takahashi et al. (1993). Why are you including TA in this recalculation? Please include accuracy and precision of Optode and Clorophyll A. The value of the one month of thermistor chain data is not properly explained. Please include reference for CO2SYS program used. There are some repetition of information that is fund in Appendix, please decide where to present it. Flux of oxygen is most commonly presented as a function of (measured O2 concentration minus saturation O2 concentration). Why not use this here?

Technical corrections:

[Figure]

P2 L25: Replace "voluntary" with "Voluntary".

P3 L10: Please rewrite sentence and include more explanation.

P4 L26: Please rewrite this line to clearly differentiate between marine station and station. E.g. "Water from the marine station, which is situated 250 m from shore, is pumped to an onshore laboratory, where analyses are performed".

P5 L15: Rewrite the paragraph starting on line 15. The temperature difference between inlet and equilibrator can be corrected for by using Takahashi et al. (1993). Why introducing the salinity-alkalinity relationship here?

P5 L26: Delete "between the".

P5 L27: Rewrite the last sentence.

P6 Ch 3.4: The chapter needs improvement/restructure.

P6 L3: Capitalize the first letter of "temperature" and "depth".

P6 L5: The depth closest to the surface was selected for what? Further, the temperature chain was deployed for only one month in 2018. Is this representative for the whole year?

P6 L8: Replace "temperature vertical profiles" with "vertical temperature profiles".

P6 L9: I suggest moving the name Ismo Willström to the Acknowledgements.

P6 L24: Replace "1) on the the" with "1) an".

P6 L26: Replace "2) on the" with "2) a".

P7 L20: Please include reference for CO2SYS.

P7 L21: Explain why you use constants from Millero (2010), as Lueker et al (2000) is recommended in open sea.

P7 L26: The TA-salininty relationship is presented in Appendix C, there is no need for
repetition, please chose where to include it.

P8 L18: Remove parenthesis. The flux of oxygen is usually presented as a function of (measured O2 concentration minus saturation concentration of O2).

P9 Figure 1 text: Please clarify in figure text from where the different data are from, e.g. in panel a) the T structure is based upon ctd station data, etc.

P9 L1: You write that r is used to describe the diurnal pCO2 variability, but in Figure 8, r is used as root mean square.

P9 L8: Replace "ca month" with either "a month" or "one month".

P10 L5: Do you mean remineralization?

P10 L11: The assumption of fresh water lenses is not introduces anywhere else, please explain more.

P12 Figure 4 text: Please replace "monthly diurnal variability" with "monthly median of diurnal variability" (as used in Chapter 5 Conclusion).

P13 L19: Rewrite the last sentence.

P14 L7: Replace "observed one during" with "observed during".

P15 L5: Replace "is an increasing trend" with "seems to be increasing throughout the day".

P16 L5: The correlation coefficient (R2) is called "r" in the panels of Figure 8. Please be consistent.

P16 L10: Please explain the sensitivity evaluation better.

P16 L11 and P17 L1: This sentence needs rewriting and more thorough explanation.

P18 L9: Replace "02" with "O2".

P18 L22: Remove "to the".

P18 L27: Replace "and smallest afternoon" with "and smallest in the afternoon".

P20 L22: Replace "solubility constant" with "solubility". Solubility is not a constant but rather dependent on T and S.

P21 L8: Hydrogen carbonate dissociate into carbonate AND hydrogen ion.

P21 L18: DIC does vary with temperature.

P22 L10: TA is highly dependent on salinity (and to a minor degree on temperature) and thus not a conservative variable.

P22 L19: "...which was based on the direct total alkalinity and salinity measurements carried out....".

P22 L25: You claim that salinity has no units, why introduce PSU?

P23: I suggest merging Appendix A: Air-sea gas exchange of CO2 and D: Gas transfer velocity, as these paragraphs discuss similar things.

P23 L9: Insert "than" between "less" and "30%".

P23 L10: Insert "a" between "purely" and "theoretical".

P23 L11: You used you own CO2 flux measurements and calculated "The best quadric fit (0.31U2....)", right? Or are you referring to Wanninkhof (1992), who presented exactly the same number? Moreover, on line 12, what is the "common parameterisation"; is it Wanninkhof (1992); 0.31U2..., or Wanninkhof (2014); 0.25U2...?

P25 L5: Please include "(ICOS)" after "Integrated Carbon Observation System".

P26 L15: The doi of this paper is placed in the middle of the word "methods". I would also recommend using the updated reference: Dickson, A.G., Sabine, C.L. and Christian, J.R. (Eds.) 2007. Guide to best practices for ocean CO2 measurements. PICES Special Publication 3, 191 pp.

---

## Author Response (AR1)

Response to Anonymous referee #1

We thank the referee for his/her comments on the article. We acknowledge that these comments helped to improve the manuscript significantly. We reply to each comment separately. The referee comments are shown with black text in italics. The response is given in blue color. The page, chapter and row numbers used here refer to the original manuscript, as the revised manuscript will not be available in the discussions section.

*General comments*

*The paper presents and discuss one year of continuous measurements of marine CO2 concentrations, gathered at the marine monitoring station of Utö island, in the northern area of the Baltic sea. Overall this article addresses a timely topic and the results presented can substantially contribute to improving estimates of the CO2 emitted and absorbed by the Baltic Sea.*

*Marine data are mainly gathered by a flow through system that pumps seawater from 5 m depth and 250 m from the coast, allowing continuous measurements of the basic physical parameters as well as dissolved oxygen, fluorescence and CO2 partial pressure (pCO2) also during harsh wintry conditions. The marine dataset is completed with direct measurements of air-sea CO2 fluxes gathered by a flux tower on the seashore and high quality atmospheric CO2 measurements acquired at the nearby meteorological station, making this dataset new and extremely valuable.*

*The data are analyzed to highlight the seasonal variability, the diurnal pCO2 cycle and to identify its main drivers. Finally, the authors estimate the bias that can be introduced calculating annual budget of the Baltic CO2 air-sea fluxes without taking into account this daily variability.*

*The main point where the paper needs a substantial improve is the overall writing and presentation of the methods and the results. I also recommend a revision of English style.*

The authors agree that the writing and presentation require polishing in order to make the manuscript more approachable. We carefully went thought the manuscript again. Additionally, as we have no native English speakers amongst the authors, the manuscript underwent a professional English proofreading.

*Methods used look generally adequate but are poorly written: you have to go back and forth the pages to understand what has been measured and where, and they often lack of detailed information on accuracy both on measured and calculated parameters. The model proposed to separate the different drivers of CO2 variability needs a better discussion mentioning also the variability that can derive from the advection of different water masses. Some trends presented in figure 6 and 7 are unexpected and need a better discussion.*

We made multiple changes in the methods section. These changes include:
- We revised the structure by dividing it into the flow-through measurements, other measurements and calculated data.
- We added more information on the measurements including accuracies.
- We added a map and short introduction of the study site.

In the results, we added discussion on the discrepancies between the modelled and observed pCO2 changes, including the effects of air-sea exchange of oxygen, mixing processes and estimation of the mixed layer depth.

More information on these updates is given in the replies below.

*The conclusions can be improved adding a comparison with other coastal areas / previous studies, where similar data are available.*

A comparison to different coastal areas was added in the conclusions section. These include the following:

P20R1:

"Multiple processes affecting the diurnal pCO2 variability at Utö were distinguished and their interplay depended on season discussed, similarly as found for the East of Gotland by Wesslander et al. (2011)"

P20R2:

"The diurnal pCO2 pattern was not clearly observed in winter time, which is in agreement to the observations in the Baltic Proper (Lanso et al., 2017)."

P20R3:

"Thus, on average, the magnitude and the timing of the diurnal pCO2 variability at Utö are similar to the ones of the pelagic conditions in the Baltic Proper, found in other studies (Wesslander et al., 2011, and Lanso et al., 2017). However, the large diurnal pCO2 signals created by upwelling indicates the stronger coastal influence at our study site."

*Specific comments*
*Please, revise the title: the paper discus the daily variability of pCO2, and related airsea*
*CO2 fluxes, not of other parameters of the CO2 system as carbonate saturation state, DIC, pH...*

The title was changed accordingly to "The diurnal cycle of the pCO2 in a coastal region of the Baltic Sea".

*Section 2 "Controls on the partial pressure of CO2"*

*This section is rather confusing. It summarizes some basic aspects of carbon chemistry and, at the same time, explains which can be neglected in the study area. I would suggest to present here the basic of carbon chemistry, with clear references to the pendix, and move the considerations on which process is important for the area, after the presentation of the study site, where the calculation performed are explained.*

Only the basics of carbon chemistry is now briefly introduced in this section. The site-specific considerations (Chapter 2.4) were moved to a new chapter, 3.8 "The included processes controlling pCO2".

*3 Materials and methods*
*Please, add a map with the location of Utö island, the station and the different sampling locations mentioned in the paper. The reference (Laasko et al 2018) includes mainly a description of the atmospheric station. This paper could be a good opportunity to present the Utö marine station with a table summarizing instrument type, variable measured, frequency etc.*

A map showing the sampling locations was added (see the next figure and its caption) in the beginning of the methods section.

[Figure]

A CTD
B Inlet
C Thermistor chain
D Marine station
E ICOS Atmospheric station
F Weather and Air quality station

Sampling locations at Utö Atmospheric and Marine Research Station. The grid size (distance between plus signs) is 1 km. The smaller figure on the upper right corner shows the location of Utö (orange star). The National Land Survey of Finland is acknowledged for providing the map.

The different marine stations are now shortly introduced in the beginning of the methods section:
"The marine observations include, but are not limited to, CTD casts carried out northwest from the island, flow-through analyses at the Marine station and thermistor measurements in the vicinity of the inlet of the seawater inlet (Fig 1)."

Also, a reference to the up-to-date observation list was added in P4R20:
"For the complete list of observations, visit Finnish Meteorological Institute's web site (https://en.ilmatieteenlaitos.fi/uto-observations). Site bathymetry and other information about the study site are given by Laakso et al. (2018) and Kraft et al. (2021)."

*The description of the instrumental setup and the whole method section is quite confusing and needs a reorganization where the description of measured data is clearly separated from calculated parameters and experimental considerations. I would suggest the following:*

*Improve the description of the study site providing general physical and biogeochemical settings of the study area. Describe the different stations where data are gathered and sampling strategy of all the data presented and discussed in the paper.*

*\* Measurements from the flow-through system: instrument used, parameters acquired, frequency, accuracy, etc.*

*\* Measurements from other stations: manual samplings, laboratory analysis, method, accuracy.*

*\* Assessment of data quality etc.. you could add a separate section or add a sentence where there is the description of data acquisition.*

*\* Calculated data: describe how you calculated all the derived parameters and estimate the uncertainty on derived carbonate system variables.*

The methods section was divided into the following sections as suggested by the reviewer: flow-through measurements, measurements from other sampling locations and calculated data. The methods section starts now with a short introduction to the different components of the marine station: a map of the sampling sites and additional text describing the sites were added. Information of the data accuracy and quality of each variable was added after the introduction of each variable.

The description of the biogeochemical setup was added in the beginning of methods section P4R14: "As the whole Baltic Sea, our study site is affected by climate change induced increase of sea water temperature (Laakso et al., 2018). Besides the warming trend, also stratification has strengthened affecting the connectivity between water layers separated by seasonal thermocline and halocline (Liblik and Lips 2019). Long-term trends in alkalinity show increase throughout the Baltic Sea, partly compensating $CO_2$-induced acidification (Müller et al, 2016). Within our study region, phytoplankton blooms are a recurrent phenomenon due to eutrophication (e.g. Kraft et al., 2021). "
The following references were added:
Liblik, T., and Lips, U.: Stratification has strengthened in the Baltic Sea – An analysis of 35 years of observational data, Front. Earth Sci., 7, 174, doi: 10.3389/feart.2019.00174, 2019.

Kraft, K., Seppälä, J., Hällfors, H., Suikkanen, S., Ylöstalo, P., Anglès, S., Kielosto, S., Kuosa, H., Laakso, L., Honkanen, M., Lehtinen, S., Oja, J., and Tamminen, T.: First application of IFCB high-frequency imaging-in-flow cytometry to investigate bloom-forming filamentous cyanobacteria in the Baltic Sea, Front. Mar. Sci., 8, 594144, doi: 10.3389/fmars.2021.594144, 2021.

*P8 The description of the calculation done is rather hard to follow. The DIC derived from mixing with other water masses is not considered in your model. Why? If you consider that this is a minor component of DIC variability in the study area, you should explain and justify your choice.*

Mixing is an important process that alters the DIC, but the estimation of its magnitude is challenging. For this reason, we added the following in the chapter that deals with the included processes controlling pCO2:

"In general, the tidal force is the most prominent process to generate a diurnal pattern on the mixing of the DIC. In this location of the non-tidal Baltic Sea, the effect of tidal currents on the water masses are

very small and thus we can neglect their effect. However, several other processes such as coastal upwelling can also generate mixing. The driving force of upwelling (or downwelling) is steady wind over the sea from suitable direction, and at our study site, close to open sea which contains very small islands, sea-breeze cannot be completely neglected but is not expected to be strong. However, there is a possibility that the density driven mixing has a diurnal cycle due to the strong diurnal pattern of solar irradiation.

The mixing component of the diurnal DIC variations can be large occasionally. For instance, there was clear indications of the mixing of water masses on July 22, 2018; the pCO2 varied 503 uatm while the water cooled by 8 °C. However, there is not always that clear indicators hinting to the mixing events. In order to analyze the effect of the mixing of DIC precisely, one has to know the 3D field of DIC and the water currents. This would require an array of carbonate system measurements, which we do not have. The analysis of the mixing of DIC is beyond the scope of this paper."

We reckon that the mismatch between the observed and modelled hourly pCO2 variations (as seen in Fig. 8) may be partly due to the effect of mixing of water masses with differing DIC content. There is a clear seasonal pattern visible in the slopes between the observed and modelled hourly pCO2 changes. For this reason, we added the following in P18R16:

"Also, our crude estimation of mixed layer depth may introduce an error in the modelled pCO2 change. It is possible that in spring, the mixing of DIC may be incomplete and the calculated mixed layer thus too deep. This causes the gas exchange of oxygen to be underestimated through Eq. 4, which further causes the biological pCO2 component to be too low. In autumn, the calculated mixed layer depth might be too shallow to represent the real mixing of carbon, as it is defined as the bottom layer depth in the cases when the mixed layer at the CTD location was found deeper than the bottom layer depth in the inlet location. It is possible that there is lateral mixing with the deeper areas. This would cause the gas exchange to be overestimated and the biological pCO2 component to be too high."

We also changed the figure 8 that shows the modeled vs observed pCO2 change to density plot that shows more clearly how the observations are distributed. This plot highlights that most observations are packed close to origo.

The most variation in our model originates from the biological component we added more discussion on the possible effects of air-sea exchange of oxygen on the biological component. The following were added in P18R16:

"It is possible that the seasonal slope changes in Fig.10 are due to the fact that the oxygen concentration change by the O2 flux is not well-described. There can be a lag between the air-sea O2 flux and the O2 concentration change at 5 m depth. It is likely that the wind speed-parameterization of O2 flux gives good estimate of the flux, but the flux at the surface is challenging to translate into the O2 concentration changes at 5 m depth at one hour resolution. In summer, the oxygen flux is directed from the sea to atmosphere, and thus its effect on the biological component should be positive. If this process is not taken into account, we might end up with an underestimated biological component, i.e. low slopes in Fig 10. In winter, vice versa would happen."

We dove little deeper in the model analysis than is shown in the manuscript. For instance, we learnt that the effect of the oxygen flux on the biological component is most important during the winter months: the change of oxygen generated by the air-sea exchange is large compared to the oxygen change. During winter months, the oxygen flux is positive and we also observe too strong modeled pCO2 component. When the oxygen flux is negative and the effect of the oxygen flux on the biological component is small in summer, we observe too low modeled pCO2 change.

The 4.2.5 chapter was slightly reorganized due to the addition of these new paragraphs. Also, the paragraph dealing with R2 values was removed as a redundant. The R2 values are still visible in the figure.

*4 Results and discussion*

*Section 4.2: why did you choose two days one after the other as an example? You could present two days from two different seasons or where the weight of the diverse pCO2 components is different. Otherwise you can show only one day of pCO2 data and add a picture with the daily variability of the other variables (temperature, oxygen,: : :).*

*Two consecutive days were selected in order to show that the variability can be large within just two days. However, the referee's suggestion is convincing as it reflects the seasonality of pCO2 diurnal cycle. We therefore now selected days from late summer (3 September) and winter (12 December). The late summer day is the same one that was in the original manuscript.*

*The following text was added on P11R10:*
*"The pCO2 on December 20, 2018, was decreasing, almost linearly (Fig. 3b). This example was chosen in order to show that the biological variation in winter is low, and the oxygen is primarily altered by mixing and air-sea exchange of oxygen. Both the air-sea exchange of carbon and cooling of water strive to decrease the pCO2."*

[Figure]

*P13: Add a description of the daily variability of temperature to improve the discussion and to allow a better comparison of your results with other areas*

We gladly followed this advice. For this reason, we calculated monthly averages of daily temperature ranges (daily maximum temperature – daily minimum temperature).

This text was added in P13R7:
"The daily variation in seawater temperature follows the cycle of solar irradiation. The highest monthly average of daily temperature ranges (daily maximum temperature – daily minimum temperature) was in July with 1.6 degC and the lowest in February with 0.2 degC."

*Both in figure 6 and 7 the data reported at h 00:00 are often markedly different from that reported at h24:00: they should be the same number or a very close one. Could you please explain better the data reported in your figures or double check your calculation?*

The time windows are not completely identical. We added the following explanation in P10R14:

"In all of the following figures, the pCO2 difference between times, t2 and t1, is plotted in the middle of t1 and t2. The first point at 00:00 represents the pCO2 difference between 00:00 and 01:00 UTC, and the last point represents the difference between 23:00 and 24:00."

*Figures*

*Figure 1: It contains a lot of information but I found it hard to read. Please, increase the size along the "y" axis. Fig 1 (a): it's hard to see the red crosses and the black line. Is this temperature from monthly CTD casts in the nearby station? You should mention it in the caption. You could also present just the top of the CTD casts, down tothe depth of the seafloor at the inlet position, and highlight the depth of the inlet to the flow-through system. Please add the conversion to local time in the caption as well.*

The size of the y axis, the red crosses and the black line were increased. The top graph shows the temperature from the CTD casts; a mention of this was added. The time zone was included in the caption.

Originally the figure 1 had five subplots and the figure 2 had only three subplots. The subplots in figure 1 were too tightly packed. For this reason, the oxygen plot was moved from figure 1 to figure 2.

[Figure]

*Figure 2: Please, increase the size of the figure and of the symbols in the captions within the figures, especially that of figure 2 (b).*

The size of the symbols in the legend of flux subplot in Fig 2 was increased. Note also that the oxygen plot was moved here, and we added the saturation concentration of the oxygen. The color coding of

the fluxes was modified in order to avoid confusion with (a) and (d) plots.

[Figure]

*Appendix D*

*The gas transfer velocity for seawater normalized at 20_C (k660) is always a positive value. The direction of the flux is given by air-sea pCO2 difference: it is directed from the sea to the atmosphere (positive) if pCO2 sea> pCO2 atm and vice-versa. When k660 is estimated from direct FCO2 measurements gathered by the eddy covariance method, the absolute value should be used. Making this correction to figure D1 the agreement between the experimental results and the theoretical model will greatly improve.*

*Thank you for pointing this out. Now, the k value is given as an absolute value, which slightly increased the R^2 value of the fit.*

[Figure]

The text was modified to take into account the absolute value of k in P23R3:
"The applicability of this parametrization for the western marine region of Utö was assessed by calculating the absolute value of k660 -"

The text was also modified to take into account new values in P23R11:
"The best quadratic fit ($0.37U10^2$) is somewhat larger than the parametrization proposed by Wanninkhof et al., 2014, which might indicate enhanced gas transfer due to the coastal characteristics of the study site. However, for best comparability and consistency, we stick with the common parametrization by Wanninkhof et al., 2014."

*Technical corrections*

*Methods:*

*P.5 L3.: which instruments log data every 15 sec? The ones mentioned at the end of the page?*

We added the sampling frequency in the description of each observation. Please see the reply for the second next question for an example.

We removed the P5R3 sentence: "Most of the instruments that analyze seawater logged data every 15 s."

*P5 L12: Is the gas calibration traceable to international CO2 standards (e.g. WMO, NOAA, ..)? If yes, please, add this information.*

The reference gases are not traceable to WMO or other international standards.

A mention was added in P5R12:
"FMI buys the reference gases from a Finnish branch of Linde-Gas (previously AGA). The gas concentrations are checked with instruments using cavity ring-down spectroscopy in FMI's laboratory prior to measurements. These instruments are calibrated using gases that are verified by the National Oceanic and Atmospheric Administration (USA). Aluminum gas containers have been used in order to minimize the concentration drift."

*P5 L15-27: There is a mix of "calculated values" with measured ones, please, describe here only measured ones. Improve the description of temperature and salinity measurements: frequency, accuracy, frequency of calibration, etc. Describe the thermistor chain when you present the marine station. Add briefly the methods for TA measurements.*

The paragraph dealing with the temperature correction of pCO2 was moved to the section of calculated data. The chapter of alkalinity-salinity relationship was removed as it was now redundant. The figure of TA-S relationship was moved to the methods section.

The following text about the thermosalinograph was added in the section dealing with the other flow-through measurements:
"The equilibrator temperature (together with salinity) was measured using a thermosalinograph (SBE45 MicroTSG, Sea-bird Scientific) that was next to the SuperCO2 instrument. The thermosalinograph is cleaned 1-2 times a year. The accuracies for temperature and salinity given by the manufacturer are respectively 0.002 °C and 0.005. The temperature drift is less than a few thousandths of a degree a year, whereas the stability of conductivity measurement depends mostly on the cleanliness of the measurement cell. The thermosalinograph logged data every 15 s."

Accuracy for the thermistors were included:
"Pt-100 thermistors were calibrated prior to the deployment in FMI's laboratory, and the maximum error in temperature was found to be less than 0.015 °C. Thermistors logged data every 30 s."

The text about total alkalinity was moved to its own chapter:
"The total alkalinity used here is calculated using a local alkalinity-salinity relationship, which is based on the samples gathered from the flow-through system at Utö in summer 2017 (Lehto, 2019). Total alkalinity was determined from these samples by using the potentiometric titration method (Metrohm Titrino 716). The samples were conserved with mercury chloride before the analysis in Finnish Environment Institute's research laboratory in Helsinki. The titrant and the rinsing water had the salinity of 7. Alkalinity was calculated from the titration curve based on the least squares method."

*P5L 28: Even if in Kilkki et al., 2015 there is a detailed description of the methods, please, add the most important details: instrument model, accuracy, sampling frequency, reference to international standards.*

We added more details on the ICOS measurement P5R28:
"The atmospheric xCO2 was measured at the Atmospheric ICOS site (Kilkki et al., 2015). The sample air was drawn from the tower (56 m) to the ground level where it was analyzed using using cavity ring-down spectroscopy (Picarro G2401). The data was logged as one minute average values. Three standard gases made by FMI were used for the reference measurement. Differences between the target and measured values of these gases were within -0.20 and 0.20 ppm. (Kilkki et al., 2015)."

*P5 L30-34: Please, add detail on instrument accuracy, frequency of calibration etc..*

More information on these sensors was added in P5R30:
"Oxygen was measured with an oxygen optode (Aanderaa 4330) with multipoint calibration. The optode has a preburned foil providing long term stability. The accuracy of the optode is 2 uM according to the manufacturer. In this paper we are mostly interested in hourly changes of oxygen, and thus the drift of absolute value is of minor concern. Chlorophyll a was measured with a Wetlabs FLNTU fluorometer, as a qualitative proxy of chlorophyll concentration, using factory calibration."

*P6 L1-12: it is not clear which data are reported in Figure 1.*

A mention of CTD was added in the caption of Fig. 1:

"Temperature of the seawater (Tw) assessed by the CTD casts – "

*P6 L8-22: move to the section of calculated parameters*

The estimation of the mixed layer depth (P6R8-22) was moved to new subsection "The determination of mixed layer depth" under the section of calculated data.

*P6 L22: delete "be"*

The excessive 'be' was removed:
"This implies that the mixed layer depths were well reproduced using the CTD castings unless the thermocline was located close to the bottom of the inlet location."

*P6 L23 Please, define "Fas" when it is used for the first time. I suggest to delete the division in two subsections or modify the titles. As they are it seems that "the flux parametrization" is related to the eddy covariance method, not a different method to calculate fluxes.*

The definition of Fas is found in P2R3.

*P7 L20 – 28: first describe how TA is calculated from salinity and the robustness of the choice done, then describe the calculations of the derived carbon chemistry parameters.*

The TA chapter was moved before introducing the CO2SYS calculations.

*P7 L29 – P8 L3: Please, rephrase and explain better.*

P7R29-P8R3 was modified:
"First, the carbon chemistry is calculated in CO2SYS for each hour based on the measured partial pressure of CO2 and the parameterized total alkalinity. This way, we derive the DIC at every hour.

In the case of the hourly temperature-related pCO2 change, we assume that DIC and TA do not change over that time frame. Using the temperature of the next hour together with the previously known DIC and TA, we calculate the new pCO2 in CO2SYS that is governed solely by the temperature change."

*P8 L4-6: Please, describe step by step, it is quite confusing*

P8R4-6 was modified to:
"In the case of air-sea exchange and biological transformations, we calculated how much DIC changed over one hour by these processes separately and added this DIC change, dDIC, to the original DIC content. Then, we calculated the carbon system using this new DIC and the unaltered total alkalinity in order to get the new pCO2."

*P8 L22: Please rephrase, it is not clear what do you mean with ": : : each day at a time".*

P8R22 was rephrased to:
"For each day, the cumulative sums of the hourly pCO2 changes generated by a specific process (temperature, biological transformations or air-sea exchange of CO2) were calculated for 00:00 – 24:00, in order to know how the specific process alters the pCO2 during a day. Finally, the mean of the cumulative sum was removed from these values, because we are interested in the daily changes, not the absolute values."

*P8 L22-26: it is not clear, what pCO'2,i is*

A definition was added in P8R24:
"pCO'2,i is the cumulative pCO2 change between the i:th and the first hour."

*P8 L28-30: please, explain better, this point is not clear.*

P8R28-30 was modified:
"This is calculated using the DIC that is altered by both the air-sea exchange of CO2 and biological transformation, and additionally taking into account the temperature change."

*Results and discussion*

*P9 L5: A more appropriate title would be "Environmental conditions and seasonal pCO2 variability".*

The title in P9R5 was changed to:
"The environmental conditions and seasonal pCO2 variability".

*Conclusions*

*P18 L30-32: All the parametrizations for gas transfer velocity contain a cubic o quadratic component function.*

The text was changed in P18R30-32 to:
"The most popular parametrizations for gas transfer velocity are either quadratic or cubic functions of the wind speed…"

Response to Anonymous referee #2

We thank the referee for his/her comments on the article. We acknowledge that these comments helped to improve the manuscript significantly. We reply to each comment separately. The referee comments are shown with black text. The response is given in blue color. The page, chapter and row numbers used here refer to the original manuscript, as the revised manuscript will not be available in the discussions section.

General comments

The paper by Honkanen et al. presents measured pCO2 data over a year from July 2018 to June 2019 from a marine site off the island Utö in the Baltic Sea. A variety of additional data is provided from the marine site, from two additional coastal sites and from the flux tower at the Utö island. The authors have examined the different drivers for the observed changes in pCO2, and they address the importance of high frequency data when precise air-sea CO2 fluxes are to be calculated. Data from coastal sites like this is highly valuable and the manuscript is very welcomed. However, the manuscript is poorly written and needs improvement both regarding English language and the structure. In the current version, the reader needs to jump back and forth to get a grip on the story, and it is not always clear what the authors want to communicate.

We followed the reviewer's comment and restructured the manuscript including specific suggestions by both reviewers. We also carefully addressed the English language and grammar. Additionally, as we have no native English speakers amongst the authors, the manuscript underwent a professional English proofreading.

Specific comments

Abstract: phrases like "monthly median diurnal pCO2 peak-to-peak amplitude" is difficult to understand. Please change to e.g. "monthly median of diurnal : : :" throughout the manuscript

All the occurrences of "monthly median diurnal pCO2 peak-to-peak amplitude" were changed to "monthly median of diurnal pCO2 variability."

This applied to P1R7, P1R11 and P2R16.

Chapter 1: The dataset are from 2018-2019, why not use a more updated reference for the global carbo cycle (e.g. the years 2009-2018: Friedlingstein et al., 2019, Earth Syst. Sci. Data, 11, 1783–1838, 2019, https://doi.org/10.5194/essd-11-1783-2019).

The reference was changed to:

Friedlingstein, P., Jones, M., O'Sullivan, M., Andrew, R., Hauck, J., Peters, G., Peters, W., Pongratz, J., Sitch, S., Le Quéré, C., Bakker, D., Canadell, J., Ciais, P., Jackson, R., Anthoni, P., Barbero, L., Bastos, A., Bastrikov, V., Becker, M., and Zaehle, S..: Global Carbon Budget 2019, Earth System Science Data, 11, 1783--1838. 10.5194/essd-11-1783-2019, 2019.

This naturally caused modifications in the values in the manuscript:

"During 2009–2018, 9.5 gigatonnes of anthropogenic carbon was released annually into the atmosphere in the form of carbon dioxide ($CO_2$) mainly through fossil fuel and land use and cement production; approximately a half of these emissions was bound by the terrestrial biosphere, 3.2 GtCy−1, and the oceans, 2.5 GtCy−1, together (Friedlingstein et al. 2019)."

Chapter 2: Please note that Appendix A is not mentioned in the text.

We added a mention of the Appendix A (P2R8):

"The partial pressure of surface seawater $CO_2$ and the direction of the air--sea $CO_2$ flux (Fas, See Appendix A)

Chapter 3 Materials and methods: This chapter needs improvement. It would be helpful if a map was included where the position of the Utö station, its water intake, thermistor chain position as well as the position of the ctd casts were marked.

A map was added in the beginning of the Materials and methods section.

[Figure]

"Figure 1. Sampling locations at Utö Atmospheric and Marine Research Station. The grid size (distance between plus signs) is 1 km. The smaller figure on the upper right corner shows the location of Utö (orange star). The National Land Survey of Finland is acknowledged for providing the map."

The marine stations were shortly introduced:
"The marine observations include, but are not limited to, CTD casts carried out northwest from the island, flow-through analyses at the Marine station and thermistor measurements in the vicinity of the inlet of the seawater inlet (Fig 1)."

We added a link to the full list of the up-to-date measurements:
"For the complete list of observations, visit Finnish Meteorological Institute's web site (https://en.ilmatieteenlaitos.fi/uto-observations)."

Also, more information on the biogeochemical and physical setup of the study were added:
""As the whole Baltic Sea, our study site is affected by climate change induced increase of sea water temperature (Laakso et al., 2018). Besides the warming trend, also stratification has strengthened affecting the connectivity between water layers separated by seasonal thermocline and halocline (Liblik and Lips 2019). Long-term trends in alkalinity show increase throughout the Baltic Sea, partly compensating $CO_2$-induced acidification (Müller et al, 2016). Within our study region, phytoplankton blooms are a recurrent phenomenon due to eutrophication (e.g. Kraft et al., 2021).
"

We updated rigorously the methods section. For the sake of clarity, we divided it into the sections of flow-through measurements, other measurements and calculated data. More information of different measurements, including accuracies, was added.

 You should clarify the difference between the marine station and station, maybe it is better to use "laboratory" for the on-shore facility?

The name of Utö Atmospheric and Marine Research Station is used for covering all of the measurements on the island. Mostly, in this paper we only use the marine station on the western edge of the island.

P4L26 was modified to:

"The marine station, located on the western shore of the island, is equipped with a flow-through system. A submersible pump located 250m from the shore transports seawater to the marine station, where seawater is analyzed automatically and manually on demand (see Fig. 1 for site map)."

"

 LI-COR 840 measures $CO_2$ and $H_2O$, but are the gas dried prior to the detection (if so, please include information). Are you using the $H_2O$ measurements to calculate dry $xCO_2$, or are you calculating $pH_2O$ from T and S?

There is a water trap on the gas line prior to the analyzer.

The following information was added in P5R10:

"Since the water trap attached to the sample gas line may slightly affect the water vapor content, the following calculation is used: the dry $CO_2$ molar fraction is calculated using the $H_2O$ measured using the analyzer. The real water vapor content in the equilibrium chambers is calculated using the temperature and salinity data assuming the full saturation. This real water vapor content is used when calculating the partial pressure of $CO_2$."

The $pCO_2$ is corrected for temperature difference between water intake and equilibrator, and this should be done according to Takahashi et al. (1993). Why are you including TA in this recalculation?

The temperature conversion factor of 0.0423 proposed by Takahashi et al.(1993) is based on measurements performed on ocean water samples, i.e. it is applicable under fully marine conditions. At the study site, salinity is below 7 and likewise all $CO_2$ system parameters reflect brackish waters conditions. Under those conditions, the actual dependence of $pCO_2$ on seawater temperature can differ significantly from the Takahashi value (See for example chapter 2 in Schneider and Müller (2018)). A temperature conversion that reflects the actual conditions at site was achieved by using standard software for $CO_2$ system calculation and TA as an additional input parameter. This is now shortly explained in the text:

"The typical temperature correction of $pCO_2$ suggested by Takahashi et al. (1993) is not applicable in the brackish conditions of the Baltic Sea."

Please include accuracy and precision of Optode and Clorophyll A.

The following information was added in P5R31:

"Oxygen was measured with an oxygen optode (Aanderaa 4330) with multipoint calibration. The optode has a preburned foil providing long term stability. The accuracy of the optode is 2 uM according to the manufacturer. In this paper we are mostly interested in hourly changes of oxygen, and thus the drift of absolute value is not concern. Chlorophyll A was measured with Wetlabs FLNTU fluorometer, as a proxy of chlorophyll concentration, using factory calibration."

The value of the one month of thermistor chain data is not properly explained.

The chain was deployed in July 2018 and was used through the entire measuring period (July 2018-June 2019).

A clarification of the data use was added in P6R7:
"The thermistor profiles were used to verify the applicability of the CTD casts carried out at slightly different location. The 3 m thermistor measurement was used for correcting the $pCO_2$ for the temperature change that occurs during the sampling procedure."

Please include reference for CO2SYS program used.

A reference for the MATLAB version was added:

"van Heuven, S., Pierrot, D., Rae, J., Lewis, E., Wallace, D.W.R.: CO2SYS v 1.1, MATLAB program developed for $CO_2$ system calculations, ORNL/CDIAC-105b, Carbon Dioxide Information Analysis Center, Oak Ridge National Laboratory, U.S. DoE, Oak Ridge, TN, 2011."

The citation was added when first time mentioning the CO2SYS in P5R16:

"Since the sample water temperature can change during the transport, we took the effect of the temperature change on $pCO_2$ into account by using the CO2SYS matlab program (van Heuven et al., 2011)."

There are some repetition of information that is fund in Appendix, please decide where to present it.

The salinity-alkalinity chapter was removed from the Appendix as redundant.

Flux of oxygen is most commonly presented as a function of (measured O2 concentration minus saturation O2 concentration). Why not use this here?

We are indeed using the concentration difference (measured O2 and calculated saturated O2 concentrations) together with wind speed dependent gas transfer velocity in order to calculate the oxygen flux.

This is now clarified in P8R18:

"This flux, FO2, is calculated similarly to the carbon dioxide flux (Eq. A1) by using the gas transfer velocity and the oxygen solubility, the measured oxygen concentration in seawater and the oxygen concentration calculated for hypothetical equilibrium with the atmosphere."

Technical corrections:

P2 L25: Replace "voluntary" with "Voluntary".

P2L25 voluntary was changed to Voluntary

P3 L10: Please rewrite sentence and include more explanation.

P3L10 and P3L14 were repetition from chapter 2.4 and thus were removed.

P4 L26: Please rewrite this line to clearly differentiate between marine station and station. E.g. "Water from the marine station, which is situated 250 m from shore, is pumped to an onshore laboratory, where analyses are performed".

The location of the different components of the infrastructure are shown in the map added, together with a clarifying text.

P4L26 was modified to:

"The marine station, located on the western shore of the island, is equipped with a flow-through system. A submersible pump located 250m from the shore transports seawater to the marine station, where seawater is analyzed automatically and manually on demand (see Fig. 1 for site map)."

P5 L15: Rewrite the paragraph starting on line 15. The temperature difference between inlet and equilibrator can be corrected for by using Takahashi et al. (1993). Why introducing the salinity-alkalinity relationship here?

P5L15 was modified:

"Since the sample water temperature can change during the transport, we took the effect of the temperature change on pCO2 into account by using the CO2SYS matlab program (van Heuven et al., 2011)."

The correction given by Takahashi et al.(1993) is based on measurements on the ocean. The Baltic Sea is a brackish sea, and for this reason we relied on using two carbon variables on the carbon system program. The salinity and alkalinity are strongly connected due to the fact that both are affected by conservative mixing.

P5 L26: Delete "between the".

P5L26 a duplicate "between the" was removed:

" -- the root mean square difference between the sea inlet and the --"

P5 L27: Rewrite the last sentence.

The sentence was rewritten:

"Because the pCO2 difference between the inlet and station is small, we conclude that the pCO2 analysis carried out at the station represents the inlet conditions."

P6 Ch 3.4: The chapter needs improvement/restructure.

The chapter was divided into smaller subsections, also addressing comments of reviewer 1: first one introduces the measurement, the second describes the mixed layer depth assessment and derivation of variables.

P6R3 was simplified:

"The thermistor chain was deployed 150 m northeast from the seawater inlet in July 2018; --"

P6 L3: Capitalize the first letter of "temperature" and "depth".

P6R3 All letters in CTD were capitalized:

"The vertical temperature profiles were measured with temperature chains, supported with regular interval profiles of Conductivity-Temperature-Depth instrument (CTD)."

P6 L5: The depth closest to the surface was selected for what?

We clarified the use of the thermistor measurement at 3 m depth:

"The 3 m thermistor measurement was used for correcting the pCO2 for the temperature change that occurs in the sampling."

Also P6R5 was modified to:

"In order to avoid instrument damages during the rough weather conditions, there was no thermistors closer than 3 m to the surface."

Further, the temperature chain was deployed for only one month in 2018. Is this representative for the whole year?

The chain was deployed in July 2018 and was used thorough the whole measuring period (July 2018-June 2019).

P6 L8: Replace "temperature vertical profiles" with "vertical temperature profiles".

P6L8 was modified accordingly:

"The mixed layer depth (zmix) was determined from the vertical temperature profiles --"

P6 L9: I suggest moving the name Ismo Willström to the Acknowledgements.

Ismo's name was moved to the acknowledgments:

"CTD profiles were taken by using a small boat, --"

P6 L24: Replace "1) on the the" with "1) an".

P6R24 the repetition was removed:

"The estimation of the air-sea exchange of $CO_2$ between the sea and atmosphere is based on two methods: (1) the eddy covariance method –"

P6 L26: Replace "2) on the" with "2) a".

P6R26 the repetition was removed:

"-- and (2) the wind speed-based flux parametrization."

P7 L20: Please include reference for CO2SYS.

The reference for CO2SYS was added

"van Heuven, S., Pierrot, D., Rae, J., Lewis, E., Wallace, D.W.R.: CO2SYS v 1.1, MATLAB program developed for CO2 system calculations. ORNL/CDIAC-105b. Carbon Dioxide Information Analysis Center, Oak Ridge National Laboratory, U.S. DoE, Oak Ridge, TN, 2011."

And citation in P7R20:

"Calculations of the carbon system were performed using the CO2SYS matlab program (van Heuven et al., 2011).

P7 L21: Explain why you use constants from Millero (2010), as Lueker et al (2000) is recommended in open sea.

Millero (2010) is developed for estuary (brackish) waters. The salinity at Utö varies between 6 and 8.

P7 L26: The TA-salininty relationship is presented in Appendix C, there is no need for repetition, please chose where to include it.

The alkalinity-salinity chapter in Appendix was removed.

P8 L18: Remove parenthesis. The flux of oxygen is usually presented as a function of (measured O2 concentration minus saturation concentration of O2).

The parenthesis around the oxygen words were removed as they were redundant in P8R18:

"This flux, FO2, is calculated similarly to the carbon dioxide flux (Eq. A1) by using the gas transfer velocity and the oxygen solubility, the measured oxygen concentration in seawater and the oxygen concentration calculated for hypothetical equilibrium with the atmosphere."

P9 Figure 1 text: Please clarify in figure text from where the different data are from, e.g. in panel a) the T structure is based upon ctd station data, etc.

A mention of CTD was added in the caption of Fig. 1:
"Temperature of the seawater (Tw) assessed by the CTD casts – "

P9 L1: You write that r is used to describe the diurnal pCO2 variability, but in Figure 8, r is used as root mean square.

The r in Fig. 8 was changed to $R^2$.

P9 L8: Replace "ca month" with either "a month" or "one month".

P9R8 "ca month" was changed to "a month":

"- --for **a** month (Fig. 2a)."

P10 L5: Do you mean remineralization?

Mineralization was changed to remineralization thorough the paper to be consistent.

P10 L11: The assumption of fresh water lenses is not introduces anywhere else, please explain more.

A clarification was added in the chapter 2.1 (P3R10):

"The processes controlling the freshwater balance evaporation, precipitation and the formation and melting of sea ice. Precipitated water or melted sea ice may produce a layer of low saline water at the sea surface, but is likely eroded easily by turbulence except under very calm conditions."

P12 Figure 4 text: Please replace "monthly diurnal variability" with "monthly median of diurnal variability" (as used in Chapter 5 Conclusion).

Fig4 caption was modified to be:

"Observed monthly medians of pCO2 diurnal variability --"

P13 L19: Rewrite the last sentence.

P13R19 was rewritten:

"The exchange of CO2 between the sea and atmosphere is driven by the CO2 partial pressure difference between them: CO2 is transported from higher partial pressure to lower. As the atmospheric partial pressure of CO2 is relatively stable compared to the one in sea, the direction is mainly given by the pCO2 in the sea. The diurnal pattern of pCO2 generated by air-sea exchange represents the accumulation of carbon in the sea in summer when the pCO2 is smaller in the sea than in the atmosphere and vice versa in winter."

P14 L7: Replace "observed one during" with "observed during".

The redundant "one" was removed in P14R7:

"-- as large as the one observed during the spring bloom."

P15 L5: Replace "is an increasing trend" with "seems to be increasing throughout the day".

P15R5 was modified accordingly:

"-- the diurnal pCO2 pattern generated by the biological processes seems to be increasing throughout the day, --"

P16 L5: The correlation coefficient (R2) is called "r" in the panels of Figure 8. Please be consistent.

Fig. 8 was updated, now with R2.

P16 L10: Please explain the sensitivity evaluation better.

P16R10 was rewritten:

"For each month, we divided the RMSE value with the average absolute change in hourly pCO2 to find out this sensitivity – "

We also did rigorous testing on the model applied. We added discussion on the possible sources of the discrepancies (air-sea exchange of oxygen, mixing, mixed layer depth) in the results.

P16 L11 and P17 L1: This sentence needs rewriting and more thorough explanation.

P16R11 was rewritten:

"Thus, the error introduced by the model during these winter months, though comparatively small in its absolute value, is large compared to the observed variability, which suggests that the estimates of the biological component during the winter time should be treated with caution."

P18 L9: Replace "02" with "O2".

The "02" was replaced with "O2":

"Wesslander et al. (2011) determined the CO2-02 ratio –"

P18 L22: Remove "to the".

The "to the" was removed:

"The atmospheric CO2 partial pressure is approximately constant when compared with the variability in the surface water."

P18 L27: Replace "and smallest afternoon" with "and smallest in the afternoon".

P18R27 was updated to:

" -- the largest difference is before noon and the smallest is in the afternoon.

P20 L22: Replace "solubility constant" with "solubility". Solubility is not a constant but rather dependent on T and S.

The "solubility constant" was replaced with "solubility".

P21 L8: Hydrogen carbonate dissociate into carbonate AND hydrogen ion.

P21R8 was corrected:

 "-- which further dissociates to carbonate (CO2– 3 ) and hydrogen ions."

P21 L18: DIC does vary with temperature.

We added the following text in P3R3 to clarify the issue:

"In contrast to pCO2, DIC and TA behave conservatively with respect to temperature changes and mixing of water masses, when expressed in concentration units of µmol per kg of seawater."

P22 L10: TA is highly dependent on salinity (and to a minor degree on temperature) and thus not a conservative variable.

We added the following text in P3R3 to clarify the issue:

"In contrast to pCO2, DIC and TA behave conservative with respect to temperature changes and mixing of water masses, when expressed in concentration units of µmol per kg of seawater."

P22 L19: ": : :which was based on the direct total alkalinity and salinity measurements carried out: : :.".

The alkalinity-salinity chapter in the Appendix was removed.

P22 L25: You claim that salinity has no units, why introduce PSU?

The alkalinity-salinity chapter in Appendix was removed, and with that, any use of PSU

P23: I suggest merging Appendix A: Air-sea gas exchange of CO2 and D: Gas transfer velocity, as these paragraphs discuss similar things.

The gas transfer velocity parametrization was merged in Appendix A as a subchapter.

P23 L9: Insert "than" between "less" and "30%".

P23R9 was corrected:

" -- fluxes can deviate less than 30% from --"

P23 L10: Insert "a" between "purely" and "theoretical".

P23R10 was corrected:

"The fully stationary condition is purely a theoretical concept, --"

P23 L11: You used you own CO2 flux measurements and calculated "The best quadric fit (0.31U2: : :.)", right? Or are you referring to Wanninkhof (1992), who presented exactly the same number? Moreover, on line 12, what is the "common parameterisation"; is it Wanninkhof (1992); 0.31U2: : :, or Wanninkhof (2014); 0.25U2: : :?

Yes, the best quadratic fit was based on our own measurements. The common parametrization means the Wanninkhof (2014), as it is the updated version. A clarification was added:

"-- we stick with the common parametrization by Wanninkhof et al. (2014)."

P25 L5: Please include "(ICOS)" after "Integrated Carbon Observation System".

The abbreviation was added:

"Also, thanks are due to the Integrated Carbon Observation System (ICOS) for providing the atmospheric CO2 data at Utö"

P26 L15: The doi of this paper is placed in the middle of the word "methods". I would also recommend using the updated reference: Dickson, A.G., Sabine, C.L. and Christian, J.R. (Eds.) 2007. Guide to best practices for ocean CO2 measurements. PICES Special Publication 3, 191 pp.

Thank you for pointing out up-to-date material. The reference was updated.

---

## Referee Report (RR1)

The page / line numbers refer to the clear version.

It is not clear where the micro-meteorological flux tower is located. Is it near "D", the Marine Station?

P7L07 "on average seawater cooled on its way from the inlet to the lab by 0.4 ± 2.0 °C" Is it correct? The variability exceeds the observed trend. In this case I would write something as: "the difference in temperature oscillates within ± 2.0°C"

P9L30 please, add temperature and salinity to the parameters used for CO2SYS calculations.

P12L08 – 12 The oxygen trend was already described at lines 1-4. I suggest to reorganize these lines to avoid repetitions that can generate confusion.

P19L08 "disabled the oxygen flux between the atmosphere and sea" What does it exactly mean? That you calculated the biological pCO2 variation assuming that all the changes in oxygen concentrations are due to production /remineralization. Is it correct?

---

## Author Response (AR2)

**OS-2020-115 Response to the anonymous referee**

The original referee comments are shown in black and the author responses in red. We thank the referee for the comments.

The page / line numbers refer to the clear version.
It is not clear where the micro-meteorological flux tower is located. Is it near "D", the Marine Station?

We added a clarification in the sentence in the line 23 in the page 3:
"The micro-meteorological flux tower at the western shore, next to the marine station, measures the $CO_2$, sensible heat and latent heat fluxes between the sea and the atmosphere."

P7L07 "on average seawater cooled on its way from the inlet to the lab by 0.4 ± 2.0 °C" Is it correct? The variability exceeds the observed trend. In this case I would write something as: "the difference in temperature oscillates within ± 2.0°C"

The variability indeed exceeds the observed trend, and thus we rewrote the sentence in the line 7 in the page 7 accordingly:
"The difference in temperature oscillates within ± 2.0 °C."

P9L30 please, add temperature and salinity to the parameters used for CO2SYS calculations.

We added the mention of temperature and salinity to the parameters used for CO2SYS in P9L30:
"First, the carbon chemistry is calculated in CO2SYS for each hour based on the measured partial pressure of $CO_2$, parameterized total alkalinity (see above), temperature and salinity."

P12L08 – 12 The oxygen trend was already described at lines 1-4. I suggest to reorganize these lines to avoid repetitions that can generate confusion.

In the lines 1-12 in the page 12, the $pCO_2$ and $O_2$ conditions are described in chronological order, starting from July 2018 and ending in June 2019. The sentence starting at P12L12 is a conclusion of the whole study period, and to emphasize this, now this sentence starts a new chapter.

We rewrote the sentence starting in P12L8 in order to emphasize that it concerns the winter:
"Also, thorough the winter, the sea was mostly a sink of oxygen and the measured oxygen concentration predominantly increased."

P19L08 "disabled the oxygen flux between the atmosphere and sea" What does it exactly mean? That you calculated the biological pCO2 variation assuming that all the changes in oxygen concentrations are due to production /remineralization. Is it correct?

That's correct, disabling the oxygen flux between the atmosphere and the sea means that all the variation in oxygen is assumed to originate from the biological transformations. We added a clarification in P19L8:

"- - we performed a similar analysis as in Fig. 9 but separately disabled the oxygen flux between the atmosphere and sea (i.e. assuming all oxygen changes to originate from the biological transformations), - -"

In addition to the changes listed above, we added an acknowledgment for the referees in P29L6:

"We thank the referees for their insightful comments resulting in highly improved manuscript."

Best regards,

Honkanen et al.